# Pre-trained Text-to-Image Diffusion Models Are Versatile Representation Learners for Control

**Gunshi Gupta**[*1]    **Karmesh Yadav**[*2]    **Yarin Gal**[1]    **Zsolt Kira**[2]    **Dhruv Batra**[2]
**Cong Lu**[1]    **Tim G. J. Rudner**[3]

[1]University of Oxford    [2]Georgia Institute of Technology    [3]New York University

## Abstract

Embodied AI agents require a fine-grained understanding of the physical world mediated through visual and language inputs. Such capabilities are difficult to learn solely from task-specific data. This has led to the emergence of pre-trained vision-language models as a tool for transferring representations learned from internet-scale data to downstream tasks and new domains. However, commonly used contrastively trained representations such as in CLIP have been shown to fail at enabling embodied agents to gain a sufficiently fine-grained scene understanding—a capability vital for control. To address this shortcoming, we consider representations from pre-trained text-to-image diffusion models, which are explicitly optimized to generate images from text prompts and as such, contain text-conditioned representations that reflect highly fine-grained visuo-spatial information. Using pre-trained text-to-image diffusion models, we construct *Stable Control Representations* which allow learning downstream control policies that generalize to complex, open-ended environments. We show that policies learned using Stable Control Representations are competitive with state-of-the-art representation learning approaches across a broad range of simulated control settings, encompassing challenging manipulation and navigation tasks. Most notably, we show that Stable Control Representations enable learning policies that exhibit state-of-the-art performance on OVMM, a difficult open-vocabulary navigation benchmark.

Code: github.com/ykarmesh/stable-control-representations

# 1   Introduction

As general-purpose, pre-trained "foundation" models [2, 5, 6, 24, 31, 34, 47] are becoming widely available, a central question in the field of embodied AI has emerged: How can foundation models be used to construct model representations that improve generalization in challenging robotic control tasks [4, 40, 64]?

Robotic control tasks often employ pixel-based visual inputs paired with a language-based goal specification, making vision-language model representations particularly well-suited for this setting. However, while vision-language representations obtained via Contrastive Language-Image Pre-training [CLIP; 33]—a state-of-the-art method—have been successfully applied to a broad range of computer vision tasks, the use of CLIP representations has been shown to lead to poor downstream performance for robotic control. This shortcoming has prompted the development of alternative, control-specific representations for embodied AI [25, 30] but has left other sources of general-purpose pre-trained vision-language representations—such as text-to-image diffusion models—largely unexplored for control applications.

---

[*]Equal Contribution.

38th Conference on Neural Information Processing Systems (NeurIPS 2024).

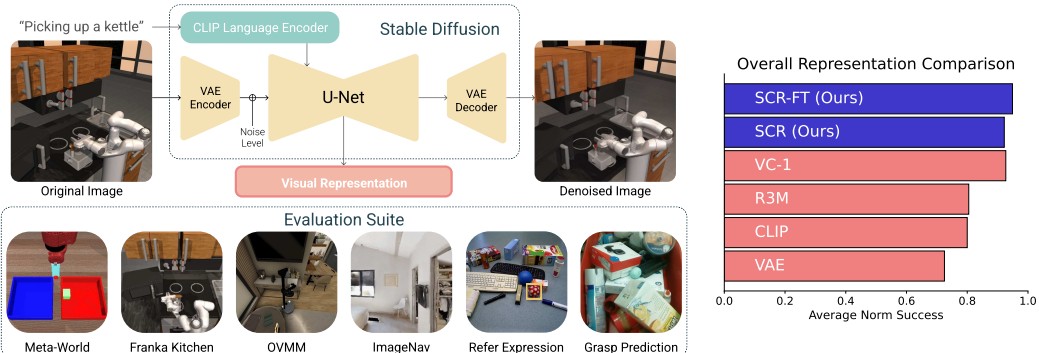

Figure 1: **Left:** Our paper proposes Stable Control Representations, which uses pre-trained text-to-image diffusion models as a source of language-guided visual representations for downstream policy learning. **Right:** Stable Control Representations enable learning control policies that achieve all-round competitive performance on a wide range of embodied control tasks, including in domains that require open-vocabulary generalization. Empirical results are provided in Section 4.

In this paper, we propose **Stable Control Representations (SCR)**: pre-trained vision-language representations from text-to-image diffusion models that can capture both high and low-level details of a scene [17, 34]. While diffusion representations have seen success in downstream vision-language tasks, for example, in semantic segmentation [3, 46, 50], they have—to date—not been used for control. We perform a careful empirical analysis in which we deconstruct pre-trained text-to-image diffusion model representations to understand the impact of different design decisions.

In our investigation, we find that diffusion representations can outperform general-purpose models like CLIP [33] across a wide variety of embodied control tasks despite not being trained for representation learning. This is the case for purely vision-based tasks as well as for settings that require task understanding through text prompts. A highlight of our results is the finding that diffusion model representations enable better generalization to unseen object categories in a challenging open-vocabulary navigation benchmark [58] and provide improved interpretability through attention maps [45].

Our key contributions are as follows:

1. In Section 3, we introduce a multi-step approach for extracting vision-language representations for control from text-to-image diffusion models. We show that these representations are capable of capturing both the abstract high-level and fundamental low-level details of a scene, offering an alternative to models trained specifically for representation learning.
2. In Section 4, we evaluate the representation learning capabilities of diffusion models on a broad range of embodied control tasks, ranging from purely vision-based tasks to problems that require an understanding of tasks through text prompts, thereby showcasing the versatility of diffusion model representations.
3. In Section 5, we systematically deconstruct the key features of diffusion model representations for control, elucidating different aspects of the representation design space, such as the input selection, the aggregation of intermediate features, and the impact of fine-tuning on performance.

We have demonstrated that diffusion models learn versatile representations for control and can help drive progress in embodied AI. Figure 1 presents a summary of our approach and results.[2]

## 2 Related Work

We begin with a review of prior work on representation learning and diffusion models for control.

**Representation Learning with Diffusion Models.** Diffusion models have received a lot of recent attention as flexible representation learners for computer vision tasks of varying granularity—ranging from key point detection and segmentation [46, 50] to image classification [48, 57]. Wang et al. [50] has shown that intermediate layers of a text-to-image diffusion model encode semantics and depth maps that are recoverable by training probes. These approaches similarly extract representations by considering a moderately noised input, and find that the choice of timestep can vary based on the granularity of prediction required for the task. Yang and Wang [57] train a policy to select an optimal

---

[2]Code link: https://github.com/ykarmesh/stable-control-representations.

diffusion timestep, we simply used a fixed timestep per class of task. Several works [45, 46, 50] observe that the cross-attention layers that attend over the text and image embeddings encode a lot of the spatial layout associated with an image and therefore focus their method around tuning, post-processing, or extracting information embedded within these layers.

**Visual Representation Learning for Control.** Over the past decade, pre-trained representation learning approaches have been scaled for visual discrimination tasks first, and control tasks more recently. Contrastively pre-trained CLIP [33] representations were employed for embodied navigation tasks by EmbCLIP [21]. MAE representations have been used in control tasks by prior works like VC-1 [27], MVP [54] and OVRL-v2 [56]. R3M [30] and Voltron [20] leverage language supervision to learn visual representations. In contrast, we investigate if powerful text-to-image diffusion models trained for image generation can provide effective representations for control.

**Diffusion Models for Control.** Diffusion models have seen a wide range of uses in control aside from learning representations. These can broadly be categorized into three areas. First, diffusion models have been used as a class of expressive models for learning action distributions for policies [7, 14, 32]; They can improve model multimodality and generate richer action distributions than Gaussians. Second, off-the-shelf diffusion models have been used to augment limited robot demonstration datasets by specifying randomizations for object categories seen in the data through inpainting [19, 28, 60]. Third, planning can be cast as sequence modeling through diffusion models [1, 11, 18].

## 3 Stable Control Representations

In this paper, we investigate the use of language-guided visual representations from the open-source Stable Diffusion model (v1.5) and their application to language-conditioned visual control tasks. We present background on latent diffusion models and text-to-image diffusion models, along with the notation we adopt in this work, in Appendix B.

To extract representations, we follow a similar protocol as Wang et al. [50], Traub [48], and Yang and Wang [57]: Given an image-text prompt, $s = \{s_{\text{image}}, s_{\text{text}}\}$, associated with a particular task, we use the SD VQ-VAE model as the encoder $\mathcal{E}(\cdot)$ and partially noise the encoded latents $z_0 \doteq \mathcal{E}(s_{\text{image}})$ to some diffusion timestep $t$, to obtain the noised latent $z_t$. We then extract a representation composed of the intermediate layer outputs of the U-Net $\epsilon_\theta$ as it produces a denoising estimate $\epsilon_\theta(z_t, t, s_{\text{text}})$. This process is illustrated in Figure 2. We refer to the extracted representations as **Stable Control Representations (SCR)**. In Sections 3.1, 3.2, 3.3, and 3.4, we describe the design space for extracting SCR, and in Sections 3.5 and 3.6, we explain how we use the representations for control.

### 3.1 Layer Selection and Aggregation

We are interested in evaluating the internal representations from the denoiser network, that is, the U-Net $\epsilon_\theta(\cdot)$. The first design choice we consider is which layers of $\epsilon_\theta$ to aggregate intermediate outputs from. The U-Net does not have a representational bottleneck, and different layers potentially encode different levels of detail. Trading off size with fidelity, we concatenate the feature maps output from the mid and down-sampling blocks to construct the representation. This results in a representation size comparable to that of the other pre-trained models we study in Section 4. This is shown at the bottom of Figure 2 and we ablate this choice in Section 5.1. Since outputs from different layers may have different spatial dimensions, we bilinearly interpolate them so that they are of a common spatial dimension and can be stacked together. We then pass them through a learnable convolutional layer to reduce the channel dimension before feeding them to downstream policies. The method used to spatially aggregate pre-trained representations can significantly affect their efficacy in downstream tasks, as we will discuss in Section 5.4. We use the best-performing spatial aggregation method for all the baselines that we re-train in Section 4.

### 3.2 Diffusion Timestep Selection

Next, we consider the choice of extraction timestep $t$ for the denoising network (shown on the left of Figure 2). Recall that the images we observe in control tasks are un-noised (i.e., corresponding to $x_0$), whereas the SD U-Net expects noised latents, corresponding to $z_t$ for $t \in [0, 1000]$. The choice of timestep $t$ influences the fidelity of the encoded latents since a higher value means more noising of the inputs. Yang and Wang [57] have observed that there are task-dependent optimal timesteps

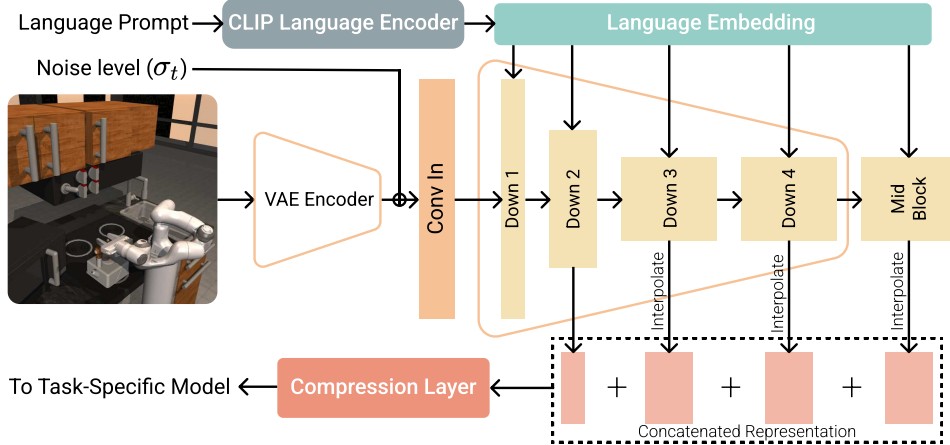

Figure 2: Extraction of Stable Control Representations from Stable Diffusion. Given an image-text prompt, $s = \{s_{\text{image}}, s_{\text{text}}\}$, we encode and noise the image and feed it into the U-Net together with the language prompt. We then aggregate feature maps from multiple layers within the U-Net, as described in Section 3. Shown here are features from the mid and downsampling blocks of the U-Net.

and proposed adaptive selection of $t$ during training, while Xu et al. [55] have used $t = 0$ to extract representations using un-noised inputs to do open-vocabulary segmentation. We hypothesize that control tasks that require a detailed spatial scene understanding would benefit from a lower diffusion timestep, corresponding to a later stage in the denoising process where the inputs have less noise. We provide evidence consistent with this hypothesis in Section 5.2. To illustrate the effect of the timestep, we display final denoised images for various $t$ values in different domains in Figure 8.

### 3.3 Prompt Specification

Since text-to-image diffusion models allow conditioning on text, we investigate if we can influence the representations to be more task-specific via this conditioning mechanism. For tasks that come with a text specifier, for example, the sentence "go to object X", we simply encode this string and pass it to the U-Net. However, some tasks are purely vision-based and in these settings, we explore whether constructing reasonable text prompts affects downstream policy learning when using the U-Net's language-guided visual representations. We present this analysis in Section 5.3.

### 3.4 Intermediate Attention Map Selection

Wang et al. [50] and Tang et al. [45] demonstrate that the Stable Diffusion model generates localized attention maps aligned with text during the combined processing of vision and language modalities. Wang et al. [50] leveraged these word-level attention maps to perform open-domain semantic segmentation. We hypothesize that these maps can also help downstream control policies to generalize to an open vocabulary of object categories by providing helpful intermediate outputs that are category-agnostic. Following Tang et al. [45], we extract the cross-attention maps between the visual features and the CLIP text embeddings within the U-Net. We test our hypothesis on an open-domain navigation task in Section 4.3, where we fuse the cross-attention maps with the extracted feature maps from the U-Net. We refer to this attention-map-augmented representation as **SCR-ATTN**.

### 3.5 Using Text-to-Image Diffusion Model Representations to Learn Control Policies

To solve visual control tasks with states given by $s = [s_{\text{image}}, s_{\text{text}}]$, where $s_{\text{text}}$ may be used to specify the task, we wish to use pre-trained vision-language representations capable of encoding the state $s$ as $f_\phi(s_{\text{image}}, s_{\text{text}})$. This encoded state is then supplied to a downstream, task-specific policy network, which is trained to predict the action $a_t$. Our evaluation encompasses both supervised learning and reinforcement learning regimes for training the downstream policies. We train agents through behavior cloning on a small set of demonstrations for the few-shot manipulation tasks we study in Section 4.1. For the indoor navigation tasks we study in Sections 4.2 and 4.3, we use a version of the Proximal Policy Optimization [PPO, 39] algorithm for reinforcement learning.

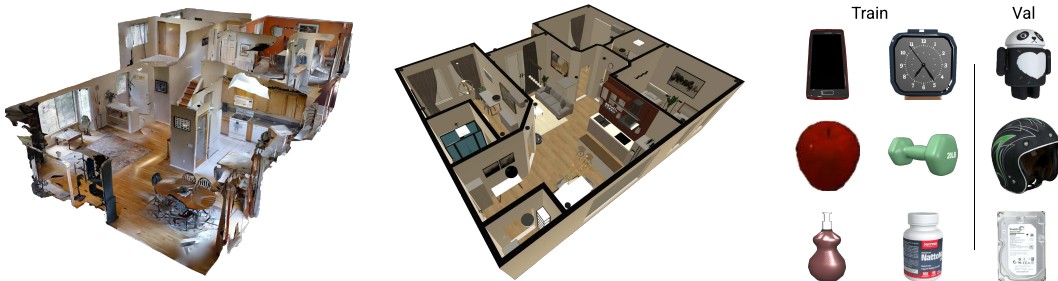

Figure 3: Sample scenes from the Habitat environments for the ImageNav (left) and OVMM (center) tasks. Instances from training and validation datasets of the OVMM object set are shown on the right.

### 3.6 Fine-Tuning on General Robotics Datasets

Finally, we consider fine-tuning strategies to better align the base Stable Diffusion model towards generating representations for control. This serves to bridge the domain gap between the diffusion model's training data (e.g., LAION images) and robotics datasets' visual inputs (e.g., egocentric tabletop views in manipulation tasks or indoor settings for navigation). Crucially, we do not require any task-specific losses for fine-tuning. Instead, we adopt the same text-conditioned generation objective as that of the base model for the fine-tuning phase. We use a small subset of the collection of datasets used by prior works on representation learning for embodied AI [27, 54]: we use subsets of the EpicKitchens [9], Something-Something-v2 [SS-v2; 13], and Bridge-v2 [49] datasets. As is standard, we fine-tune the denoiser U-Net $\epsilon_\theta$ but not the VAE encoder or decoder. Image-text pairs are uniformly sampled from the video-text pairs present in these datasets. A possible limitation of this strategy is that text-video aligned pairs (a sequence of frames that correspond to a single language instruction) may define a many-to-one relation for image-text pairs. However, as we see in experiments in which we compare to the base Stable Diffusion model in Section 4, this simple approach to robotics alignment is useful in most cases. Further details related to fine-tuning are provided in Appendix E.1. We refer to the representations from this fine-tuned model as **SCR-FT**.

## 4 Empirical Evaluation

In this work, we evaluate Stable Control Representations (SCR) on an extensive suite of tasks from 6 benchmarks covering few-shot imitation learning for manipulation in Section 4.1, reinforcement learning-based indoor navigation in Sections 4.2 and 4.3, and tasks related to fine-grained visual prediction in Appendices D.2 and D.3. Together, these tasks allow us to comprehensively evaluate whether our extracted representations can encode both high and low-level semantic understanding of a scene to aid downstream policy learning. We describe the common baselines used across tasks in Appendix C, and present the individual task setups and results in the following subsections.

### 4.1 Few-shot Imitation Learning

We start by evaluating SCR on commonly studied representation learning benchmarks in few-shot imitation learning. Specifically, our investigation incorporates five commonly studied tasks from Meta-World [59] (same as CORTEXBENCH [27]), which includes bin picking, assembly, pick-place, drawer opening, and hammer usage; as well as five tasks from the Franka-Kitchen environments included in the RoboHive suite [23], which entail tasks such as turning a knob or opening a door. We adhere to the training and evaluation protocols adopted in their respective prior works to ensure our results are directly comparable (detailed further in Appendix G.1).

**Results.** We report the best results of SCR and baselines in Table 1a. On Meta-World, we see that SCR outperforms most prior works, achieving 94.9% success rate. In comparison, VC-1, the visual foundation model for embodied AI and CLIP achieved 92.3 and 90.1% respectively. On Franka-Kitchen, SCR obtains 49.9% success rate, which is much higher than CLIP (36.3%) and again outperforms all other baselines except for R3M. We note that R3M's sparse representations excel in few-shot manipulation with limited demos but struggle to transfer beyond this setting [27, 20]. We see that while the SD-VAE encoder performs competitively on Franka-Kitchen, it achieves a low success rate on Meta-World. This observation allows us to gauge the improved performance

Table 1: Average Success Rate and standard error evaluated across different representations.

(a) Meta-World & Franka-Kitchen.

| Model | Meta-World | Franka-Kitchen |
|---|---|---|
| R3M | **96.0 ± 1.1** | **57.6 ± 3.3** |
| CLIP | 90.1 ± 3.6 | 36.3 ± 3.2 |
| VC-1 | 92.3 ± 2.5 | 47.5 ± 3.4 |
| Voltron | 72.5 ± 5.2 | 33.5 ± 3.2 |
| SD-VAE | 75.5 ± 5.2 | 43.7 ± 3.1 |
| SCR | **94.4 ± 1.9** | 45.0 ± 3.3 |
| SCR-FT | **94.9 ± 2.0** | 49.9 ± 3.4 |

(b) ImageNav

| Model | Success |
|---|---|
| R3M | 30.6 |
| CLIP-B | 52.2 |
| VC-1 | 70.3 |
| MVP | 68.1 |
| SD-VAE | 46.6 |
| SCR | **73.9** |
| SCR-FT | 69.5 |

(c) OVMM

| Model | Success |
|---|---|
| Oracle | 77.6 |
| Detic | 36.7 |
| CLIP | 38.7 ± 1.7 |
| VC-1 | 40.6 ± 2.2 |
| SCR | 38.7 ± 1.2 |
| SCR-FT | **41.9 ± 1.0** |
| SCR-FT-ATTN | **43.6 ± 2.1** |

of SCR from the base performance gain we may get just from operating in the latent space of this VAE. Additionally, we see that the task-agnostic fine-tuning gives SCR-FT an advantage (4%) over SCR on Franka-Kitchen while making no difference on Meta-World. Note that the other high-performing baselines (R3M and Voltron) have been developed for downstream control usage with training objectives that take temporal information into account, while VC-1 has been trained on a diverse curation of robotics-relevant data. In this context, SCR's comparable performance shows that generative foundation models hold promise for providing useful representations for control, even with relatively minimal fine-tuning on non-task-specific data.

## 4.2 Image-Goal Navigation

We now assess SCR in more realistic visual environments, surpassing the simple table-top scenes in manipulation benchmarks. In these complex settings, the representations derived from pre-trained foundational models are particularly effective, benefiting from their large-scale training. We study Image-Goal Navigation (ImageNav), an indoor visual navigation task that evaluates an agent's ability to navigate to the viewpoint of a provided goal image [63]. The position reached by the agent must be within a 1-meter distance from the goal image's camera position. This requires the ability to differentiate between nearby or similar-looking views within a home environment. This task, along with the semantic object navigation task that we study in Section 4.3, allows for a comprehensive evaluation of a representation's ability to code both semantic and visual appearance-related features in completely novel evaluation environments. We follow the protocol for the ImageNav task used by Majumdar et al. [27] and input the pre-trained representations to an LSTM-based policy trained with DD-PPO [52] for 500 million steps on 16 A40 GPUs (further details in Appendix G.3). Given the large compute requirements for training, we directly compare SCR and SCR-FT to the results provided in Majumdar et al. [27].

**Results.** We evaluate our agent on 4200 episodes in 14 held-out scenes from the Gibson dataset and report the success rate in Table 1b. We find that SCR outperforms all other representations, while the fine-tuned version SCR-FT is almost on par with the second-best-performing VC-1 (69.5% vs 70.3%), the SOTA visual representation from prior work. This can be expected given that it was fine-tuned on images solely from table-top manipulation datasets. We also see that R3M, the best model for few-shot manipulation from Table 1a performs very poorly (30.6%) in this domain, showing its limited transferability to navigation tasks.

## 4.3 Open Vocabulary Mobile Manipulation

We now shift our focus to evaluating how Stable Diffusion's web-scale training can enhance policy learning in open-ended domains. We consider the Open Vocabulary Mobile Manipulation (OVMM) benchmark [58] that requires an agent to find, pick up, and place objects in unfamiliar environments. One of the primary challenges here is locating previously unseen object categories in novel scenes (illustrated in Figure 3 (left)). To manage this complex sparse-reward task, existing solutions [58] divide the problem into sub-tasks and design modular pipelines that use open-vocabulary object detectors such as Detic [62] to enable generalization to novel objects. We study a modified version of the Gaze sub-task (detailed in Appendix G.2), which focuses on locating a specified object category for an abstracted grasping action.

Table 2: We analyze the impact of varying the denoising timestep, layers selection, and input text prompt for the performance of SCR on the Franka-Kitchen benchmark. We report the mean and standard error over 3 random seeds.

| (a) Denoising timestep. | |
| --- | --- |
| Timestep | Success Rate |
| 0 | **49.9 ± 3.4** |
| 10 | **48.2 ± 3.1** |
| 100 | 42.0 ± 3.7 |
| 110 | 42.0 ± 3.4 |
| 200 | 35.1 ± 3.2 |

| (b) Layers selection. | |
| --- | --- |
| Layers | Success Rate |
| Down[1-3] + Mid | **49.9 ± 3.4** |
| Down[1-3] | 43.0 ± 3.4 |
| Mid | 41.6 ± 3.3 |
| Mid + Up[0] | 42.1 ± 3.6 |
| Mid + Up[0-1] | **48.1 ± 3.6** |

| (c) Input text prompt. | |
| --- | --- |
| Prompt Type | Success Rate |
| None | **49.9 ± 3.4** |
| Relevant | 49.2 ± 3.5 |
| Irrelevant | 48.7 ± 3.3 |

The task's success is measured by the agent's ability to precisely focus on the target object category. This category is provided as an input to the policy through its CLIP text encoder embedding. The evaluation environments cover both novel instances of object categories seen during policy learning, as well as entirely unseen categories. We compare to VC-1, the best model from Section 4.2 and CLIP, since prior work has studied it for open-vocab navigation [21, 26]. We also incorporate a baseline that trains a policy with access to ground truth object masks, evaluated using either the ground truth or Detic-generated masks at test time (labeled as Oracle/Detic).

**Results.** Table 1c shows that SCR-FT surpasses VC-1 by 1.3%, beating CLIP and SCR by 3.2%. It is surprising that VC-1's visual representation does better than CLIP's image encoder representation, given that the downstream policy has to use these with the CLIP text encoder's embedding of the target object category. Comparing these to SCR-FT-ATTN, we can see the benefit of providing intermediate outputs in the form of text-aligned attention maps to the downstream policy (+1.7%). Samples of attention maps overlaid on images from an evaluation episode can be found in Appendix G. These word-level cross-attention maps simultaneously improve policy performance and also aid explainability, allowing us to diagnose successes and failures. Interestingly, the foundation model representations (CLIP, VC-1, SCR) perform better than Detic. While object detections serve as a category-agnostic input for downstream pick-and-place policies, noisy detections can often lead to degraded downstream performance, as we see in this case. Nonetheless, there is still a sizeable gap to 'Oracle' which benefits from ground truth object masks at test-time.

## 5 Deconstructing Stable Control Representations

In this section, we deconstruct Stable Control Representations to explain which design choices are most determinative of model robustness and downstream performance.

### 5.1 Layer Selection

We begin our investigation by examining how the performance of SCR is influenced by the selection of layers from which we extract feature maps. We previously chose outputs from the mid and downsampling layers of the U-Net (Figure 2), because their aggregate size closely matches the representation sizes from the ViT-based models (VC-1, MVP, and CLIP). Appendix E.2 details the feature map sizes obtained for all the models we study. Table 2a lists the success rates achieved on the Franka-Kitchen domain when we use different sets of block outputs in SCR. We present similar ablations for Meta-World in the top four rows of Table 3b.

We observe that utilizing outputs from multiple layers is instrumental to SCR's high performance. This finding underscores a broader principle applicable to the design of representations across different models: Leveraging a richer set of features from multi-layer outputs should enhance performance on downstream tasks. However, it is important to acknowledge the practical challenges in applying this strategy to ViT-based models due to the high dimensionality of each layer's patch-wise embeddings (16×16×1024 for ViT-L for images of size 224×224). We present the success rates achieved on the four benchmarks when aggregating multi-layer embeddings from CLIP models in Tables 3a and 4, alongside SCR (the representation size for which is now half in comparison). In Table 3a, we observe that moving towards middle layers leads to higher performance indicating that CLIP layers 10-14 encode some details useful to the Franka-Kitchen benchmark. While we see benefits from including the output of certain additional layers, it still underperforms SCR.

Table 3: Layer-selection ablations across different benchmarks.

(a) Ablations for CLIP on Franka-Kitchen.

| Model | Layers | Success |
|---|---|---|
| CLIP-L | 23 (last layer) | 36.3 ± 1.7 |
| CLIP-L | 21+23 | 35.4 ± 2.9 |
| CLIP-L | 19+23 | 38.5 ± 3.2 |
| CLIP-L | 12+23 | 40.8 ± 2.8 |
| CLIP-L | 10+23 | 40.2 ± 3.2 |
| SCR (ours) | Down[1-3] + Mid | **49.9 ± 3.4** |

(b) Ablations for SCR on Meta-World.

| Layers | Noise | Success |
|---|---|---|
| Mid | 200 | 94.7 ± 2.8 |
| Down[3] + Mid | 200 | **97.3 ± 1.4** |
| Down[1-3] | 200 | 94.1 ± 1.9 |
| Down[1-3] + Mid | 200 | 94.4 ± 1.9 |
| Down[1-3] + Mid | 100 | 94.4 ± 1.9 |
| Down[1-3] + Mid | 0 | 94.1 ± 1.9 |

Table 4: Comparison of CLIP Layer Ablations on Meta-World, OVMM, and ImageNav

(a) Meta-World and OVMM

| Model | Layers | Meta-World | OVMM |
|---|---|---|---|
| CLIP-L | 23 (Last Layer) | 90.1 ± 3.6 | 38.7 ± 1.7 |
| CLIP-L | 21+23 | 91.2 ± 2.3 | - |
| CLIP-L | 12+23 | 91.7 ± 2.6 | 38.6 ± 1.6 |
| SCR | Down[1-3] + Mid | **94.9 ± 2.0** | **43.6 ± 2.1** |

(b) ImageNav

| Model | Layers | ImageNav |
|---|---|---|
| CLIP-B | 11 (Last Layer) | 52.2 |
| CLIP-B | 6+11 | 66.6 |
| SCR | Down[1-3] + Mid | **73.9** |

## 5.2 Sensitivity to the Noising Timestep

Next, we characterize the sensitivity of task performance to the denoising step values chosen during representation extraction. We present results on the Franka-Kitchen tasks in Table 2b, and on the Meta-World tasks in the bottom three rows of Table 3b. We see that the performance across nearby timesteps (0 and 10 or 100 and 110) is similar, and that there is a benefit to doing a coarse grid search up to a reasonable noising level (0 vs 100 vs 200) to get the best value for a given task.

## 5.3 How is Language Guiding the Representations?

Recall that in the OVMM experiments (Section 4.3), we concatenated the target object's CLIP text embedding to the visual representations before feeding it to the policy. For SCR and SCR-FT, we also provided the category as the text prompt to the U-Net, and additionally extracted the generated cross-attention maps for SCR-FT-ATTN. In this subsection, we seek to more closely understand how the text prompts impact the representations in SCR.

We first consider the Franka-Kitchen setup from Section 4.1, which includes manipulation tasks that do not originally come with a language specification. We experiment with providing variations of task-relevant and irrelevant prompts during the representation extraction in SCR. Table 2c shows the downstream policy success rates for irrelevant (*"an elephant in the jungle"*) and relevant (*"a Franka robot arm opening a microwave door"*) prompts, compared to the default setting of not providing a text prompt We see that providing a prompt does not help with downstream policy performance and may even degrade performance as the prompt gets more irrelevant to the visual context of the input.

We now move to the Referring Expressions Grounding task which requires predicting a bounding box for an object being referred to in a sentence, within an image depicting cluttered objects. We defer the main presentation of this task to Appendix D.2 and use it to probe the degree of language grounding in SCR in this section. To study the role of the U-Net in shaping the visual representations guided by the text, we examine different text integration methods to generate SCR representations in Table 5.

We compared the following approaches for providing the task's text specification to the task decoder (also depicted in Figure 4):

(a) **No text input:** Exclude text prompt from both SCR and the task decoder by passing an empty prompt to the U-Net and using only the resulting SCR output for the decoder.
(b) **Prompt only:** Pass text prompt only to the U-Net.
(c) **Concat only:** Concatenate the CLIP embedding of the text prompt with the visual representation, feeding an empty prompt to the U-Net.
(d) **Prompt + Concat:** Combine "Prompt Only" and "Concat Only".
(e) **Only text encoding:** Ignore the visual representation and rely only on CLIP text embeddings.

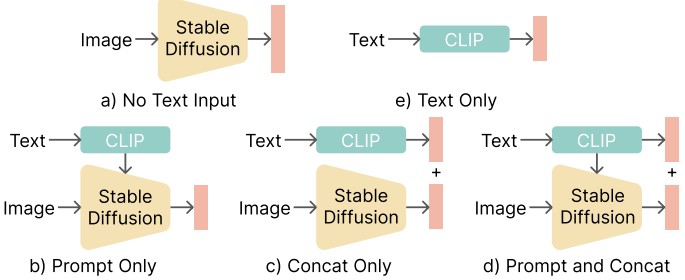

a) No Text Input

e) Text Only

b) Prompt Only

c) Concat Only

d) Prompt and Concat

Figure 4: Illustration of different approaches to providing relevant vision-language inputs to a downstream task-decoder.

Table 5: Ablating text input to SCR on the referring expressions grounding task.

| Configuration | Score |
|---|---|
| (a) No text input | 14.8 |
| (b) Prompt only | 82.7 |
| (c) Concat only | **92.2** |
| (d) Prompt + Concat | **92.9** |
| (e) Only text encoding | 37.5 |

Investigating the results of (a) and (b) in Table 5, it is evident that incorporating the text prompt into the U-Net significantly enhances accuracy compared to ignoring the text altogether. The difference in scores between (b) and (c) indicates that directly providing text embeddings to the decoder improves performance, suggesting that certain crucial aspects of object localization are not fully captured by the representation alone. Comparing (c) to (d), we see that with concatenated text embeddings, further modulation of the visual representations does not provide significant benefits. Finally, the significant decrease in the score for (e) reveals the extent to which the task relies on text-based guesswork.

These findings align with both intuition and recent research on controllable generation with diffusion models [61] that highlights the challenges associated with using long-form text guidance. There are ongoing research efforts, focused on training models with more detailed descriptions or leveraging approaches to encode and integrate sub-phrases of long texts, that seek to address these challenges.

## 5.4 The Effect of Spatial Aggregation

In this study, we refine the approach for extracting representations by integrating a convolutional layer that downsamples the spatial grid of pre-trained representations. This adjustment, referred to as a "compression layer" by Yadav et al. [56], aims to reduce the high channel dimension of pre-trained model outputs without losing spatial details, facilitating more effective input processing by downstream task-specific decoders.

We explore the effect of spatial aggregation methods by comparing the convolutional downsampling layer method to multi-headed attention pooling (MAP) used for CLIP embeddings in Karamcheti et al. [20]. We find that using a compression layer significantly improves performance on the fine-grained visual prediction tasks described in Appendix D as reported in Table 6 (columns 3-4). This result challenges the conjecture made in prior work that CLIP representations are limited in their ability to provide accurate low-level spatial information [20] and emphasizes the critical role of appropriate representation aggregation.

Building on this result, we assess whether better spatial aggregation can improve the performance of CLIP representations on downstream control tasks. We present these results in Table 6 (columns 5-6) for VC-1 and CLIP on the MuJoCo tasks. We see that the compression layer often outperforms the use of CLS token embeddings (by 1-2%), but CLIP representations still fail to match the best-

Table 6: We ablate the spatial aggregation method for VC-1 and CLIP. On the fine-grained visual prediction tasks, we compare the average precision between using multi-head attention pooling (MAP) and the compression layer. On the Meta-World & Franka-Kitchen tasks, we compare the average success rates ($\pm$ one standard error) between the CLS token and compression layer embeddings.

| Model | Aggregation Method | Refer Exp. Grounding | Grasp Affordance Prediction | Meta-World | Franka-Kitchen |
|---|---|---|---|---|---|
| VC-1 | MAP/CLS | 93.2 | 24.7 | 88.8 $\pm$ 2.2 | **52.0 $\pm$ 3.4** |
| VC-1 | Compression | **94.6** | **83.9** | **92.3 $\pm$ 2.5** | 47.5 $\pm$ 3.4 |
| CLIP | MAP/CLS | 68.1 | 60.3 | 88.8 $\pm$ 3.9 | 35.3 $\pm$ 3.4 |
| CLIP | Compression | **94.3** | **72.9** | **90.1 $\pm$ 3.6** | **36.3 $\pm$ 3.2** |

performing models. This result provides evidence that the underperformance of CLIP representations on control tasks is unlikely due to a lack of sufficiently fine-grained visual information. Finally, we note that the compression layer aggregation technique was used for all baselines in Tables 1b and 1c to ensure a strong baseline comparison. We recommend that future studies adopt this methodology to enable a fairer comparison of representations.

## 6  Discussion

In Section 5, we deconstructed Stable Control Representations and highlighted techniques used in our approach that could be applied to extract representations from other foundation models. Our analysis in Sections 5.1 and 5.4 revealed that using multi-layer features and appropriate spatial aggregation significantly affects performance, and overlooking these factors can lead to misleading conclusions about the capabilities of previously used representations.

Next, our investigation into how language prompts shape diffusion model representations uncovered nuanced results and showed that text influence on representations does not consistently increase downstream utility. This is particularly evident in tasks where text specification is not required and where training and test environments are congruent, minimizing the need for semantic generalization. In contrast, tasks like referring expressions grounding demonstrate the necessity of direct access to text embeddings for accurate object localization, even when representations are modulated to considerable success. For the OVMM task, we identified a scenario where multimodal alignment is essential and proposed a method to explicitly utilize the latent knowledge of the Stable Diffusion model through text-aligned attention maps, which is not straightforward to do for other multimodal models. Future research could design methods to derive precise text-associated attribution maps for other models.

Finally, we contrasted the simplicity of fine-tuning diffusion models with that of the contrastive learning objective required to fine-tune CLIP representations. The former only requires image-text samples for the conditional generation objective, whereas the latter requires a sophisticated negative label sampling pipeline along with large batch sizes to prevent the model from collapsing to a degenerate solution [33]. We demonstrate this phenomenon empirically on the Franka-Kitchen environment by fine-tuning CLIP similarly to SCR-FT in Appendix D.1.

## 7  Conclusion

In this paper, we proposed Stable Control Representations, a method for leveraging representations of general-purpose, pre-trained diffusion models for control. We showed that using representations extracted from text-to-image diffusion models for policy learning can improve generalization across a wide range of tasks including manipulation, image-goal and object-goal based navigation, grasp point prediction, and referring expressions grounding. We also demonstrated the interpretability benefits of incorporating attention maps extracted from pre-trained text-to-image diffusion models, which we showed can improve performance and help identify downstream failures of the policy during development. Finally, we discussed ways in which the insights presented in this paper, for example, regarding feature aggregation and fine-tuning, may be applicable to other foundation models used for control. We hope that Stable Control Representations will help advance data-efficient control and enable open-vocabulary generalization in challenging control domains as the capabilities of diffusion models continue to improve.

## Acknowledgments

GG is funded by the EPSRC Centre for Doctoral Training in Autonomous Intelligent Machines and Systems (EP/S024050/1) and Toyota Europe. We gratefully acknowledge donations of computing resources by the Alan Turing Institute. The Georgia Tech effort was supported in part by ONR YIP and ARO PECASE. The views and conclusions contained herein are those of the authors and should not be interpreted as necessarily representing the official policies or endorsements, either expressed or implied, of the U.S. Government, or any sponsor.

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

# Appendix

**Table of Contents**

## A Wider Societal Impacts

This paper presents work whose goal is to advance the field of robotic control using pretrained foundation models. Since we primarily aim to learn policies for simulated control tasks, we do not anticipate any ethical concerns at this stage. However, when deploying foundation models in the real world, we should be wary that our policies do not inherit any structural biases from the pretraining stage or internet-scale data.

## B Background

### B.1 Diffusion Models

Diffusion models [16, 42] are a class of generative models that learn to iteratively reverse a forward noising process and generate samples from a target data distribution $p(\boldsymbol{x}_0)$, starting from pure noise. Given $p(\boldsymbol{x}_0)$ and a set of noise levels $\sigma_t$ for $t = 1, \ldots, T$, a denoising function $\epsilon_\theta(\boldsymbol{x}_t, t)$ is trained on the objective

$$\mathcal{L}_{\mathrm{DM}}(\theta) = \mathbb{E}_{\boldsymbol{x}_0, \epsilon, t}[\|\epsilon - \epsilon_\theta\left(\boldsymbol{x}_t, t)\right)\|_2^2] = \mathbb{E}_{\boldsymbol{x}_0, \epsilon, t}[\|\epsilon - \epsilon_\theta\left(\boldsymbol{x}_0 + \sigma_t \cdot \epsilon, t)\right)\|_2^2], \qquad (\mathrm{B.1})$$

where $\epsilon \sim \mathcal{N}(0, 1)$, $t \sim \mathrm{Unif}(1, T)$, and $\boldsymbol{x}_0 \sim p(\boldsymbol{x}_0)$. To generate a sample $\boldsymbol{x}_0$ during inference, we first sample an initial noise vector $\boldsymbol{x}_T \sim \mathcal{N}(0, \sigma_T)$ and then iteratively denoise this sample for $t = T, ..., 1$ by sampling from $p(\boldsymbol{x}_{t-1}|\boldsymbol{x}_t)$, which is a function of $\epsilon_\theta(\boldsymbol{x}_t, t)$.

In some settings, we may want to generate samples with a particular property. For example, we may wish to draw samples from a conditional distribution over data points, $p(\boldsymbol{x}_0|c)$, where $c$ captures some property of the sample, such as classification label or a text description [34, 36]. In these settings, we may additionally train with labels to obtain a conditioned denoiser $\epsilon_\theta(\boldsymbol{x}_t, t, c)$ and generate samples using classifier-free guidance [15].

### B.2 Latent Diffusion Models

Latent diffusion models [34] reduce the computational cost of applying diffusion models to high-dimensional data by instead diffusing low-dimensional representations of high-dimensional data. Given an encoder $\mathcal{E}(\cdot)$ and decoder $\mathcal{D}(\cdot)$, Equation (B.1) is modified to operate on latent representations, $\boldsymbol{z}_0 \doteq \mathcal{E}(\boldsymbol{x}_0)$, yielding

$$\mathcal{L}_{\mathrm{LDM}}(\theta) = \mathbb{E}_{\boldsymbol{x}_0, c, \epsilon, t}[\|\epsilon - \epsilon_\theta\left(\mathcal{E}(\boldsymbol{x}_0) + \sigma_t \cdot \epsilon, t, c)\right)\|_2^2], \qquad (\mathrm{B.2})$$

where $\epsilon \sim \mathcal{N}(0, 1)$, $t \sim \mathrm{Unif}(1, T)$, $\boldsymbol{x}_0, c \sim p(\boldsymbol{x}_0, c)$. After generating a denoised latent representation $\mathbf{z}_0$, it can be decoded as $\boldsymbol{x}_0 = \mathcal{D}(\boldsymbol{z}_0)$.

A popular instantiation of a conditioned latent diffusion model is the text-to-image Stable Diffusion model [SD; 34]. The SD model is trained on the LAION-2B dataset [38] and operates in the latent space of a pre-trained VQ-VAE image encoder [12]. The model architecture is shown at the top of Figure 1 and is based on a U-Net [35], with the corresponding conditioning text prompts encoded using a CLIP language encoder [33].

### B.3 Policy Learning for Control

We model our environments as Markov Decision Processes (MDP, Sutton and Barto [43]), defined as a tuple $M = (\mathcal{S}, \mathcal{A}, P, R, \gamma)$, where $\mathcal{S}$ and $\mathcal{A}$ denote the state and action spaces respectively, $P(s'|s, a)$ the transition dynamics, $R(s, a)$ the reward function, and $\gamma \in (0, 1)$ the discount factor. Our goal is to optimize a policy $\pi(a|s)$ that maximizes the expected discounted return $\mathbb{E}_{\pi, P}\left[\sum_{t=0}^{\infty} \gamma^t R(s_t, a_t)\right]$.

In this paper, we consider visual control tasks that may be language-conditioned, that is, states are given by $s = [s_{\mathrm{image}}, s_{\mathrm{text}}]$, where $s_{\mathrm{text}}$ specifies the task. We are interested in pre-trained vision-language representations capable of encoding the state $s$ as $f_\phi(s_{\mathrm{image}}, s_{\mathrm{text}})$. This encoded state is then supplied to a downstream, task-specific policy network, which is trained to predict the action $a_t$. Our evaluation encompasses both supervised learning and reinforcement learning regimes for training the downstream policies. We train agents through behavior cloning on a small set of demonstrations for the few-shot manipulation tasks we study in Section 4.1. For the indoor navigation tasks we study in Sections 4.2 and 4.3, we use a version of the Proximal Policy Optimization [PPO, 39] algorithm for reinforcement learning.

## C  Baselines

We compare SCR and its variants (i.e., SCR-FT and SCR-FT-ATTN) to the following prior work in representation learning for control:

1. **R3M** [30] pre-trains a ResNet50 encoder on video-language pairs from the Ego4D dataset using time-contrastive video-language alignment learning.
2. **MVP** [54] and **VC-1** [27] both pre-train ViT-B/L models with the masked auto-encoding (MAE) objective on egocentric data from Ego4D, Epic-Kitchens, SS-v2, and ImageNet, with VC-1 additionally pre-training on indoor navigation videos.
3. **CLIP** [33] trains text and ViT-based image encoders using contrastive learning on web-scale data.
4. **Voltron** [20] is a language-driven representation learning method that involves pre-training a ViT-B using MAE and video-captioning objectives on aligned text-video pairs from SS-v2.
5. **SD-VAE** [34] is the base VAE encoder used by SD to encode images into latents.

To assess how well the vision-only methods would do on tasks with language specification, we concatenate their visual representations with the CLIP text embeddings of the language prompts. While we are limited by the architecture designs of the released models we are studying, to ensure a more fair comparison we try to match parameter counts as much as we can. We use the ViT-Large (307M parameters) versions of CLIP, MVP, and VC-1 since extracting SCR involves a forward pass through 400M parameters.

## D  Further Empirical Results

### D.1  Fine-tuning CLIP

We follow the same experimental constraints that we took into account while fine-tuning the diffusion model to get SCR-FT: we trained it on the same text-image pairs from the same datasets and used CLIP's contrastive loss to bring the visual embedding of the middle frames of a video closer to the video caption's text embedding. Specifically, for our experiment, we use the huggingface CLIP finetuning implementation and train the model with a batch size

Table 7: Performance on Franka-Kitchen after fine-tuning CLIP.

| Model | Franka-Kitchen |
| --- | --- |
| CLIP | 36.9 ± 3.2 |
| CLIP (FT) | 34.2 ± 2.9 |

of 384 (the maximum number of samples we were able to fit on 8 A40 GPUs) with a learning rate of 5e-5 and a weight decay of 0.001 for 5000 update steps (same as SR-FT). We present the results in Table 7 for Franka-Kitchen, and note the lack of improvement on the task post-fine-tuning.

### D.2  Referring Expressions Grounding

In Sections 4.1 to 4.3, our analysis focused on the performance of various representations across an array of control tasks. We now turn our attention to two downstream tasks involving fine-grained visual prediction. The first task, Referring Expressions Grounding, is detailed within this section, while the second task, Grasp Affordance Prediction, is discussed in Appendix D.3. Karamcheti et al. [20] have previously examined the performance on these tasks as proxy measures to evaluate the efficacy of representations for control applications.

The **Referring Expressions Grounding** task requires the identification and bounding box prediction of an object in an image based on its textual description. Similar to Karamcheti et al. [20], we use the OCID-Ref Dataset [51] for our experiments. We show a sample image-text pair from the dataset to showcase the complexity of the task in Figure 5. The frozen visual representation is concatenated with a text embedding and passed to a 4-layer MLP, which predicts the bounding box coordinates.

**Results.** We report the bounding box accuracy at a 25% Intersection-over-Union (IoU) threshold across different scene clutter levels for SCR-variants and baselines in Table 8. Firstly, we present the results from [20] for CLIP, R3M and Voltron and observe that Voltron outperforms the other two baselines. We see that SCR is tied with Voltron and that VC-1 and SD-VAE perform the best with a 1.5% lead. The better performance of these vision-encoder-only methods highlights that on this task, it is not a challenge for the downstream decoder to learn to associate the visual embeddings with the (CLIP) text encoding of the language specification. Since the training budget is fixed, we observed

that some of the runs could potentially improve over extended training. However, we were primarily interested in this task not just to compare the downstream visual prediction performance, but to use it as a testbed for exploring the following two questions: (1) Do the performance differences between the representations we evaluated in Sections 4.1 to 4.3, stem from the absence of fine-grained spatial information encoded within the representations? We refute this claim in Section 5.4, where we present the impact of the representations' spatial aggregation method on prediction performance. (2) Additionally, we explore to what extent language prompting influences the representations from SCR on language-conditioned tasks in Section 5.3.

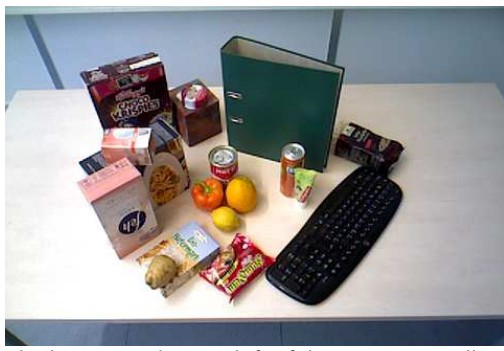

The lemon on the rear left of the instant_noodles.

Figure 5: Sample from the OCID-Ref dataset used for the Referring Expressions task.

| Model | Average | Maximum clutter | Medium clutter | Minimum clutter |
|---|---|---|---|---|
| CLIP | 68.1 | 60.3 | 76.6 | 67.0 |
| R3M | 63.3 | 55.3 | 68.3 | 63.3 |
| Voltron | 92.5 | 96.9 | 91.8 | 90.2 |
| VC-1 | 94.6 | 93.7 | 96.5 | 93.7 |
| SD-VAE | 94.3 | 93.2 | 96.3 | 93.4 |
| SCR | 92.9 | 91.1 | 95.9 | 91.8 |
| SCR-FT | 91.8 | 90.1 | 94.8 | 90.8 |

Table 8: Referring Expression Grounding (Accuracy at threshold IoU of 0.25 with label.).

### D.3 Grasp Affordance Prediction

In this section, we present our experiments on a second visual prediction task continuing from the experiments in Section 4.1. The **Grasp Affordance Prediction** task requires predicting per-pixel segmentation outputs for certain areas of objects in an RGB image. These areas correspond to parts of the surface that would be amenable to grasping by a suction gripper. The evaluation metric adopted in prior work is the precision of predictions corresponding to positive graspability at varying confidence levels (90, 95, and 99th percentile of the predicted per-pixel probabilities, denoted as Top90, Top95, and Top99 in Table 9). We refer the reader to Karamcheti et al. [20] for the complete task setup details.

Table 9: Grasp Affordance Prediction: Precision on pixels corresponding to positive graspability at varying probability threshold levels.

| Model | Top99 | Top95 | Top90 |
|---|---|---|---|
| CLIP | 60.3 | 45.0 | 28.6 |
| CLIP (Comp) | 72.9 | 55.9 | 36.5 |
| Voltron | 62.5 | 42.8 | 32.1 |
| SD-VAE | 55.6 | 41.3 | 33.8 |
| SCR | 72.8 | 55.9 | 54.5 |
| SCR-FT | 72.3 | 54.6 | 44.4 |

We re-ran all the methods using the evaluation repository provided with the work, and obtained different results compared to the reported numbers in Karamcheti et al. [20], which we attribute to a bug that we fixed related to the computation of the precision metrics. The evaluation procedure for this task adopted in prior work involves a 5-fold cross-validation, and we observed a high variability in the results, with different runs of 5-fold cross-validation yielding different final test metrics. Our findings highlight that SCR and our adaptation of CLIP (in gray, detailed in Section 5.4) both excel at this task, achieving a Top99 score of 72.9. Interestingly, we see that fine-tuning did not enhance the performance of SCR on the visual prediction tasks explored in this section and Section 4.1, suggesting a potential disconnect between visual prediction and control task benchmarks.

### D.4 Comparison with LIV

We include a comparison with LIV [25] on two tasks that involve manipulation and navigation. LIV is a vision-language representation learned through contrastive learning on the EpicKitchens dataset [9]. Similar to R3M results in the main paper, this representation does well on manipulation tasks but poorly on navigation tasks.

Table 10: Comparing to LIV on manipulation and navigation tasks.

| Model | Franka-Kitchen | OVMM |
|-------|----------------|------|
| SCR | 45.0 | 38.7 |
| SCR-FT | 49.9 | 41.9 |
| LIV | 54.2 | 8.4 |

### D.5 Overall Ranking of Representations

In Table 11, we present the consolidated scores across the four control benchmarks we study in Section 4, for all the representations we evaluate in this work. This is to give a higher-level view of the all-round performance of the different representations on the diverse set of tasks we consider. We see that VC-1, SCR, and SCR-FT emerge as the top three visual representations overall. While VC-1 is a representation-learning foundation model trained specifically for robotics tasks, SCR and SCR-FT are the diffusion model representations that we study in this paper, confirming the potential of large pre-trained foundation generative models across a wide array of downstream robotics tasks.

Table 11: Representation Performance Comparison: Numbers in the task columns (OVMM, Image-Nav, MetaWorld, Franka Kitchen) indicate relative scores of different representations (normalized by the highest score on that task), and the average normalized score column indicates the averaged scores across the task-wise relative scores where numbers are available.

| Method | OVMM | ImageNav | MetaWorld | Franka Kitchen | Average Norm. Score |
|--------|------|----------|-----------|----------------|---------------------|
| VAE | - | 0.629 | 0.786 | 0.759 | 0.725 |
| R3M | - | 0.414 | 1.000 | 1.000 | 0.805 |
| VC-1 | 0.969 | 0.951 | 0.961 | 0.825 | 0.927 |
| CLIP | 0.924 | 0.706 | 0.939 | 0.630 | 0.800 |
| SR | 0.924 | 1.000 | 0.983 | 0.781 | 0.922 |
| SR-FT | 1.000 | 0.942 | 0.989 | 0.866 | **0.949** |

# E Implementation Details

## E.1 Fine-tuning Stable Diffusion

We used the `runwayml/stable-diffusion-v1-5` model weights provided by Huggingface to initialize our models and fine-tuned them using the *diffusers* library[3]. As noted in Section 3.6, we used a subset of the frames from the EpicKitchens, Something-Something-v2, and Bridge-v2 datasets. We extracted the middle-third of the video clips and sampled four frames randomly from this chunk to increase the chances of sampling frames where the text prompt associated with the video clip is most relevant for describing the scene. Since the Something-Something-v2 (SS-v2) dataset includes a human hand manipulating table-top objects while the Bridge-V2 dataset includes a robot gripper, we append the text "human hand" and "robot hand" respectively to the text caption associated eith the respective frames to reduce ambiguity in the image-text pairing. Using this procedure, we generated a paired image-language dataset with 1.3 million samples. Figure 6 shows samples of the images from the fine-tuning datasets. Since different embodiments (human and robot) are visible in the training images, we prepended the corresponding embodiment name to the text prompt for the associated image during training.

We fine-tuned for only a single epoch ($\sim$5,000 gradient steps) parallely on 2 Nvidia A100 GPUs with a mini-batch size of 512 and a learning rate of $1e^{-4}$. Although the original Stable Diffusion model is trained on images of resolution $512 \times 512$, we fine-tuned the model on images downscaled to $256 \times 256$ since it aligned with the resolution requirements of the downstream application. We show sample generations from the diffusion model after fine-tuning in Figure 7. We found that the model learns to associate the prompt with not just the human or robot hand but also with the style of the background and objects of the training datasets.

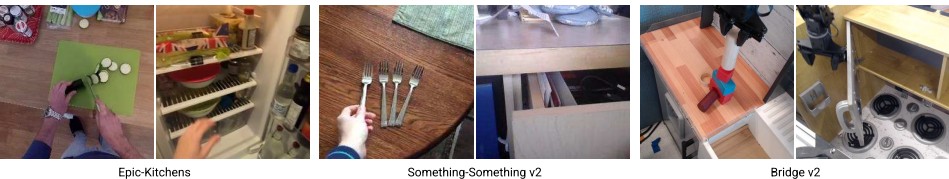

Figure 6: Snapshots from the datasets we use for fine-tuning the Stable Diffusion model.

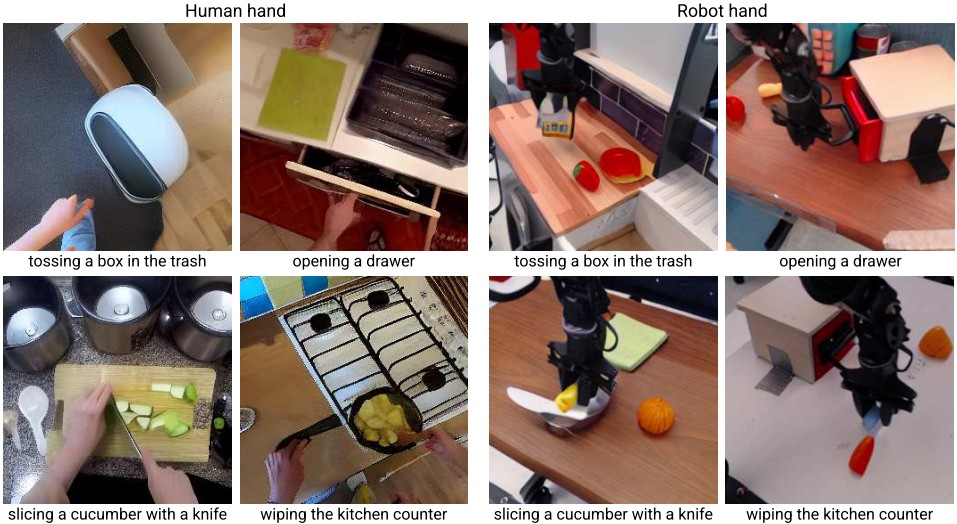

Figure 7: Image generations from the fine-tuned Stable Diffusion model. We provided four different prompts, each prefixed with either *"Human hand"* or *"Robot hand"*.

---

[3]https://huggingface.co/docs/diffusers/v0.13.0/en/training/text2image

### E.2 Representation Extraction Details

Here, we describe the representation extraction details for all our baselines assuming a $224 \times 224$ input image.:

1. **Stable Control Representations**: The Stable Diffusion model downsamples the input images by a factor of 64. Therefore, we need to first resize the input image to a size of $256 \times 256$. We pass the image to the VAE, which converts it into a latent vector of size $32 \times 32 \times 4$ and passes it to the U-Net. We use the last three downsampling blocks' and the mid block's output feature map of sizes $8 \times 8 \times 640$, $4 \times 4 \times 1,280$, $4 \times 4 \times 1,280$, and $4 \times 4 \times 1,280$, respectively. The total size is, therefore, $102,400$, and we linearly interpolated them to the same spatial dimension ($8 \times 8$) before concatenating them channel-wise.
2. **R3M** [30]: For most of our experiments we use the original ResNet50 model, which outputs a 2048 dimensional vector. For the referring expressions and grasp affordance prediction tasks from the Voltron evaluation suite [20], a VIT-S is used, which outputs an embedding of size $14 \times 14 \times 384 = 75,264$
3. **MVP** [54] and **VC-1** [27]: The last layer ($24^{\text{th}}$) outputs an embedding of size $16 \times 16 \times 1,024 = 262,144$.
4. **CLIP** [33]: For ViT-B, the last layer ($12^{\text{th}}$) outputs an embedding of size $14 \times 14 \times 768 = 150,528$. For ViT-L, the last layer ($24^{\text{th}}$) outputs an embedding of size $16 \times 16 \times 1024 = 262,144$.
5. **Voltron** [20]: We use the VCond-Base model which outputs a representation of size $14 \times 14 \times 768 = 150,528$.
6. **SD-VAE** [34]: Outputs a latent vector of size $32 \times 32 \times 4 = 4,096$.

### E.3 Hyperparameters

We provide the hyperparameters used in Section 4 for Stable Control Representations in Table 12.

Table 12: Hyperparameters and configuration settings used across tasks and methods.

| Benchmark | Timestep | Prompt | Attn | Layers | Post Compression Dim |
|---|---|---|---|---|---|
| Meta-World | 0/100/200 | No | No | Mid + Down [1-3] | 3072 |
| Franka-Kitchen | 0 | No | No | Mid + Down [1-3] | 2048 |
| ImageNav | 0 | No | No | Mid + Down [1-3] | 2048 |
| OVMM | 100 | Yes | Yes | Mid + Down [1-3] | 2048 |
| Referring Expression | 0 | Yes | No | Mid + Down [1-3] | 8192 |
| Grasp Prediction | 0 | No | No | Mid + Down [1-3] | 8192 |

## F   Runtime Analysis

We include a runtime analysis for SCR versus VC-1 in Table 13 by reporting the time taken for a forward pass through both models over a single input image of size $256 \times 256$. We use an A100 GPU for our experiment, and use both the models at half precision. We see that SCR takes 0.021 seconds per inference step, being 1.5x slower than the forward pass through VC-1 Large.

Table 13: Runtimes for.

| | SCR | VC-1 |
|---|---|---|
| Time per forward pass (Averaged over 1,000 passes) | 0.021 seconds | 0.014 seconds |

# G  Task Descriptions

## G.1  Few-Shot Imitation Learning

For all baselines, we freeze the pre-trained vision model and train a policy using imitation learning on the provided set of 25 expert demonstrations. The results are then reported as the average of the best evaluation performance for 25 evaluation runs over 3 seeds. All experiments are conducted on a single A100 GPU with 24 CPUs and 188 GBs of RAM.

**Meta-World.** We follow Majumdar et al. [27] and use the hammer-v2, drawer-open-v2, bin-picking-v2, button-press-topdown-v2, assembly-v2 tasks from the Meta-World benchmark suite [59]. Each task provides the model with the last three $256 \times 256$ RGB images, alongside a 4-dimensional gripper pose. The model consists of a 3-layer MLP with a hidden dimension of 256, utilizing ReLU as the activation function. It undergoes training for 100 epochs, with evaluations conducted every 10 epochs, following the approach of Majumdar et al. (2023). The training uses a mini-batch size of 256 and a learning rate of $10^{-3}$.

**Franka-Kitchen.** The tasks involved here include Knob On, Knob Off, Microwave Door Open, Sliding Door Open, and L Door Open, each observed from three distinct camera angles. For each task, the model receives a $256 \times 256$ RGB image and a 24-dimensional vector representing the manipulator's proprioceptive state. For our experiments, we use the the RoboHive Kumar et al. [23] GitHub repository[4] and use a 2-layer MLP with a hidden dimension of 256 and train for 500 epochs. The mini-batch size is set at 128, with a learning rate of $10^{-4}$. *We additionally correct a bug in the RoboHive implementation of the VC-1 baseline, specifically on input image normalization. Adjusting the image normalization to a 0-1 range resulted in a significant improvement in its performance.*

## G.2  Open Vocabulary Mobile Manipulation

Open-Vocabulary Mobile Manipulation [OVMM; 58] is a recently proposed embodied AI benchmark that evaluates an agent's ability to find and manipulate objects of novel categories in unseen indoor environments. Specifically, the task requires an agent to "*Find and pick an* `object` *on the* `start_receptacle` *and place it on the* `goal_recetacle`", where `object`, `start_receptacle` and `goal_recetacle` are the object category names. Given the long-horizon and sparse-reward nature of this task, current baselines [58] divide the problem into sub-tasks, which include navigation to the start receptacle, precise camera re-orientation to focus on the object (an abstracted form of grasping), navigating to the goal receptacle, and object placement.

Since our aim is to investigate the open-vocabulary capabilities of pre-trained representations, we choose to evaluate the models on only the precise camera re-orientation task (more commonly known as the **Gaze** task). In the original Gaze task, the agent is initialized within a distance of 1.5m and angle of $15°$ from the `object` which is lying on top of the `start_receptacle`. The episode is deemed successful when the agent calls the `Pick` action with the camera's center pixel occupied by the target object and the robot's gripper less than 0.8m from the object center. In our initial experiments, we found the current initialization scheme would lead the agent to learn a biased policy. This policy would call the `Pick` action after orienting towards the closest object in the field of view. Therefore, we chose to instantiate a harder version of the gaze task, where the episode starts with the agent spawned facing any random direction within 2.0m of the object.

We carry out our experiments using the HomeRobot GitHub repository[5]. HomeRobot uses the Habitat simulator [44] with the episode dataset provided by Yenamandra et al. [58]. This dataset uses 38 scenes for training and 12 scenes for validation, all originating from the Habitat Synthetic Scenes Dataset [HSSD; 22]. The episode dataset populates the scenes with extra objects from other datasets including Amazon Berkeley Objects [ABO; 8], Google Scanned Objects [ABO; 10] and Alfred [41]. The validation scenes are populated with objects not seen during training, spanning 106 seen and 22 unseen categories. The validation set consists of a total of 1199 episodes.

Our agent is designed to resemble the Stretch robot, characterized by a height of 1.41 meters and a radius of 0.3 meters. At a height of 1.31 meters from the base, a 640x480 resolution RGBD camera is mounted. This camera is equipped with motorized pan and tilt capabilities. The agent's action space

---

[4] https://github.com/vikashplus/robohive
[5] https://github.com/facebookresearch/home-robot

is continuous, allowing it to move forward distances ranging from 5 to 25 centimeters and to turn left or right within angles ranging from 5 to 30 degrees. Additionally, the agent can adjust the head's pan and tilt by increments ranging from 0.02 to 1 radian in a single step.

In our experiments, we use a 2 layer LSTM policy and pass in the visual encoder representations after passing them through the compression layer. We initialize the LSTM weights with the LSTM weights of the Oracle model to get a slight boost in performance. We train our agents using the distributed version of PPO [52] with 152 environments spread across 4 80GB Nvidia A100 GPUs. Each run also has access to 96 CPUs and 754 GBs of RAM. We train for 100M environment steps while evaluating the agent every 5M steps and report the metrics based on the highest success rate observed on the validation set.

### G.3   ImageNav

We conduct our ImageNav experiments in the Habitat simulator [37], using the episode dataset from Mezghani et al. [29]. These experiments are conducted using the VC-1 codebase[6] [27]. The dataset uses 72 training and 14 validation scenes from the Gibson [53] scene dataset with evaluation conducted on a total of 4200 episodes. The agent is assumed to be in the shape of a cylinder of height 1.5m and radius 0.1m, with an RGB camera mounted at a height of 1.25m from the base. The RGB camera has a resolution of 128×128 and a $90°$ field-of-view.

At the start of each training episode, an agent is randomly initialized in a scene and is tasked to find the position from where the goal image was taken within 1000 simulation steps. At each step, the agent receives a new observation and is allowed to take one of the four discrete actions including MOVE_FORWARD (25cm), TURN_LEFT ($30°$), TURN_RIGHT ($30°$) and STOP. The episode is a success if the agent calls the STOP action within 1m of the goal viewpoint. Similar to [56, 27] we train our agents using a distributed version of DD-PPO [52] with 320 environments for 500M timesteps (25k updates). Our experiments are conducted using 2 nodes containing 8 A40 GPUs each with a total of 128 CPUs and 504 GBs of RAM. Each environment accumulates experience across up to 64 frames, succeeded by two PPO epochs using two mini-batches. While the pre-trained model is frozen, the policy is trained using the AdamW optimizer, with a learning rate of $2.5 \times 10^{-4}$ and weight decay of $10^{-6}$. We use the reward function proposed by [56]. Performance is assessed every 25M training steps, with reporting metrics based on the highest success rate observed on the validation set.

---

[6]https://github.com/facebookresearch/eai-vc

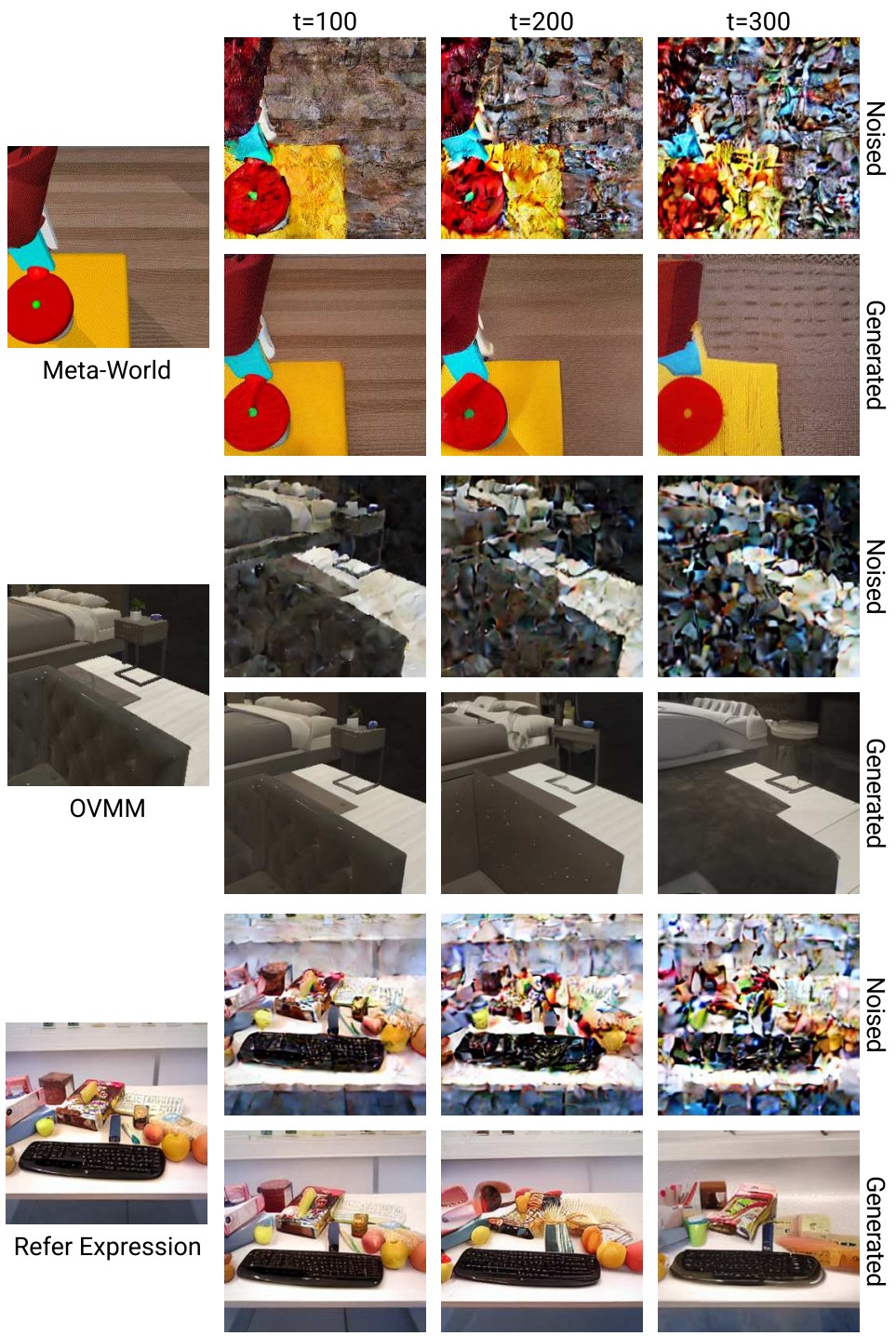

Figure 8: Noising and denoising plots for images from 3 of our tasks using the fine-tuned Stable Diffusion model. For each image, we first add noise up to timestep $t$, where $t \in \{100, 200, 300\}$, and then denoise the image back to timestep 0. We observe that inputs from different tasks are differently sensitive to the noising ranges, based on the amount of information the images contain. On Meta-World, SD is able to reconstruct the image correctly even at $t = 200$, while for the referring expressions grounding task, noising leads to information loss even at $t = 100$ with several small objects reconstructed differently to the original. This affects the range of the noise we could add to the ended latents at the time of representation extraction. It should be noted however, that using un-noised inputs ($t = 0$) always worked well in our experiments, and this hyper-parameter could be ignored in practice.

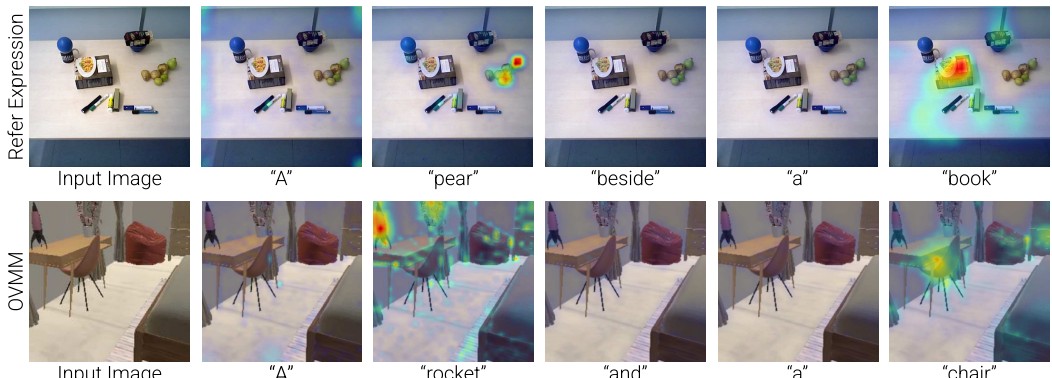

Figure 9: The Stable Diffusion model allows us to extract word-level cross-attention maps for any given text prompt. We visualize these maps in a robotic manipulation environment and observe that they are accurate at localizing objects in a scene. Since these maps are category agnostic, downstream policies should become robust to unseen objects at test time.

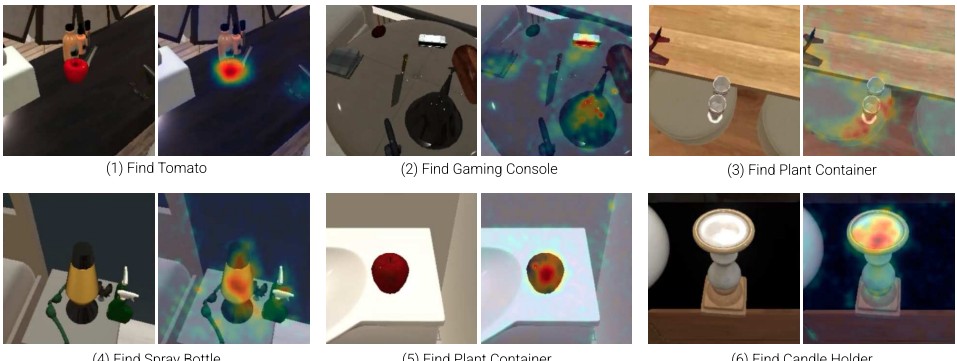

Figure 10: Images from OVMM benchmark with their corresponding attention maps obtained from the fine-tuned Stable Diffusion (SD) model. The first 5 pairs of images correspond to failed episodes, with the bottom right pair corresponding to a successful episode. The attention maps help us interpret the cause of failure: (1) Tomato - SD wrongly attends strongly to an apple. (2) Gaming Console - visible at the top of the image; however, SD attends to multiple objects due to low visual quality. (3) Plant Container - SD instead focuses on the two glasses it sees in the image. (4) Spray Bottle - SD completely misses the spray bottles in the image and attends to the lava lamp. (5) Plant Container - SD wrongly attends to the apple. (6) Candle Holder - SD correctly attends.

