# OpenReview forum: "Pre-trained Text-to-Image Diffusion Models Are Versatile Representation Learners for Control"
_NeurIPS.cc/2024/Conference — NeurIPS 2024 spotlight_

### Official Review · Reviewer_zJpU · 2024-06-26

**Soundness:** 4
**Presentation:** 4
**Contribution:** 3
**Rating:** 8
**Confidence:** 3

**Summary:**

The research question is whether representation from image generation models are superior to other pre-training paradigms (i.e. contrastive CLIP) for embodied/navigation-type of tasks. The main intuition given is that these tasks require a lot of fine-grained (e.g. spatial) understanding. While CLIP has been used as a foundation model in traditional vision or some vision+language tasks, it has not shown much promise for robotic control.
Their method is similar to previous works and each design choice is ablated/explained : First a certain timestep (i.e. noise level) is chosen for denoising a given image with an associated text prompt. Then a concatenation of intermediate U-Net layer activations is used as the “Stable Control Representation” (SCR).
They evaluate their method on three different types of robotic-related tasks and show that SCR is overall the strongest, albeit not the strongest on every single task. For example, R3M is slightly stronger on the few-shot manipulation tasks (Meta World, Franka Kitchen) but is drastically worse on navigation.

**Strengths:**

This is the first paper to make Diffusion Model representations work for robotic control tasks (novelty).
The work is also situated clearly in the previous literature context and the technical background is well explained. Similarly, the paper is well structured and very easy to follow.
Each part of the paper is well motivated and answers a natural question or follow-up step.

Regarding the technical details, it is rigorous work that actually advances science and makes it easy for others to use for their research, e.g. in the appendix it is mentioned: “We re-ran all the methods using the evaluation repository provided with the work, and obtained different results compared to the reported numbers in Karamcheti et al. [20], which we attribute to a bug that we fixed related to the computation of the precision metrics.”.
Or similarly they note in the appendix that the original setup for the Gaze task leads to a trivial solution and modify the task setup. This shows the authors don’t just blindly adopt tasks but critically investigate details of all the data or tasks they chose.

The authors use baselines, i.e. showing SD-VAE is an insightful baseline that is easily beat.
Similarly, ensuring fair and scientific baselines comparison i.e. in l. 327 - 329 where baselines are made stronger with insights from the paper to make it fair.

Lastly, it is great to see an anonymous code repo that seems to be done well and with the aim of reproducibility.

**Weaknesses:**

Only minor improvements, hard to beat some baselines properly on first glance. But on second glance it seems like stronger general representations, some of the other baselines such as R3M are only useful on some tasks but drastically drop on others. Maybe for people not familiar with this exact domain, you could explain more how these baselines work and why they get the numbers they do? For example, I am a vision-and-language person who is also familiar with all the text-to-image modeling advances but not so much with the navigation/control details.

It is briefly acknowledged in section 5.1. but imo requires a deeper investigation: is the success primarily from fusing multiple layers and not from the fact that it is a diffusion model vs. a CLIP-like model? The authors acknowledge that such an approach is less feasible with i.e. ViT since the feature maps are much larger but nonetheless this question seems important enough to nonetheless investigate further.

Overall the paper is very well written but if I had to pick a confusing section, it is 5.3 where too many things are introduced at once.

The last paragraph in the discussion about the simplicity of finetuning SCR compared to CLIP comes out of nowhere and should be discussed earlier, and the claim about sophisticated negative sampling only cites the original CLIP paper which afaik only has random negative samples? There is an experiment in the appendix that somewhat supports this but also does not discuss the negative sampling. Not a major issue but should be made clearer.

**Questions:**

Why cite Imagen in l. 36?
Reference to Table 2a/b in section 5.1 and 5.2 are mixed up, just a small fix.

“bilinearly interpolate them so that they are of a common spatial dimension and can be stacked together”: this should be specified or shown in equations formally to understand exactly what is done. Is this method common for extracting/unifying internal representations?

**Limitations:**

There are not many concerns to address and the authors wrote an appropriate short section in the Appendix.

---

> ### Author Rebuttal · Authors · 2024-08-07
>
> Thank you for taking the time to review our manuscript and for providing helpful and constructive feedback. We were happy to see your review mention the novelty of our work and the rigorousness and fairness of our evaluation. We address your concerns below.
>
> ---
>
> **1) The mutliple layer results for CLIP should be investigated further.**
>
> **We agree and include the results for this in the general comment.** On a high level, our conclusions from this experiment are that we do see a benefit from including additional outputs from certain layers along with the last layer which is always used (as we anticipated in the discussion in sec 5.1). We find these layers using a grid search, and they are located towards the middle (10-14th layers). We note that **while the feature map sizes after including another layer for CLIP are very large and not comparable anymore to the other methods, the performance still doesn’t match that of SCR**.
>
> **2) Could you explain more how these baselines work and why they get the numbers they do?**
>
> We agree with the reviewer and will add further contextualization of the performance of different baselines.
> - In the manipulation space, representation learning works adopt a few shot imitation learning setting, with the backbone pretrained on video datasets of humans manipulating objects in indoor environments.
> - In the navigation space, the setting is large scale reinforcement learning, and the evaluation includes generalization to new objects or scenes. The backbone is pretrained on datasets of indoor navigation videos.
> - We select some of the best performing baselines used in prior works in manipulation (R3M, MVP) or navigation (VC-1) tasks. Different baselines often do well on disparate sets of tasks and settings. Specifically, R3M representations are tailored to reduce overfitting in few-shot IL settings by forcing sparsity in representations (see Table 1 in the R3M paper where performance drops 6% when the L1 sparsity penalty is removed), which doesn’t offer an advantage in the extended training regime of RL-based navigation tasks.
>
> **3) Simplicity of finetuning SCR compared to CLIP should be discussed earlier, claim about sophisticated negative sampling only cites the original CLIP paper**
>
> We agree that the discussion about simplicity of finetuning SD along with the CLIP fine-tuning experiment appears late and are happy to push it into the main paper. We cited the CLIP paper for the use of really large batches sizes in contrastive learning, but a more detailed list of references would be the following:
> - References for "contrastive learning needs large batch sizes to avoid learning degenerate solutions" : the CLIP and SimCLR papers which rely on in-batch negatives, as well as momentum contrast methods like MoCo which cache mini-batch computations and use them in later batches.
> - References for "Mining hard negatives can avoid having to use large batch sizes (but is itself non-trivial)" : “Contrastive Learning with Hard Negative Samples” [2], “Debiased Contrastive Learning” [3].
>
> We did not use hard negative mining in our experiments and simply use the largest batch size which we could fit across 8x46GB A40 GPUs for CLIP finetuning.
>
> **4) Sec 5.3 where too many things are introduced at once.**
>
> We agree that section 5.3 abruptly introduces a lot of new things, and are happy to take the reviewer’s suggestion on the presentation of this section. We had wanted to present the following three results and had to choose a subset to retain in the main paper due to space constraints:
> - We wanted to introduce the proxy visual grounding tasks used in Voltron [20] and present results for SCR on them.
> - We wanted to ablate the method of spatial aggregation for CLIP and other baselines and show that retaining spatial information changes the conclusions of [20] completely on these tasks. We further wanted to show that a similar gain (or at least as considerable) is not seen on the actual control tasks.
> - We further used the above visual grounding tasks to ablate how we provide the language inputs to SCR in the case of language guided tasks, and assess to what extent language modulates the representation.
>
> **5) Bilinear interpolation should be shown in equations formally to understand exactly what is done**
>
> We will modify figure 2 in the paper to include the equation along with the visual description of this concatenation operation. We primarily do this **to enable stacking of the differently sized U-net feature maps so that we can use a single convolutional layer to process them without adding any other learnable parameters**. A similar resizing operation was most recently used in [1], which looks at training a feature adapter module for control.
>
> **6) a) Why cite Imagen in L36? b) Reference to Table 2a/b are mixed up.**
>
> Thanks for pointing these out. a) We will move this reference (meant to be a citation for examples of foundation diffusion models) to the background instead, and b) fix the swapped references.
>
> ---
>
> We hope that these clarifications and additional experimental results have addressed any remaining questions and concerns.
>
> If you find that we have addressed your questions and concerns, we kindly ask you to consider raising your score to reflect that. If issues still remain, then we would be more than happy to answer these in the discussion period. Thank you.
>
> ---
>
> **References**
>
> [1] Lin, Xingyu, et al. "Spawnnet: Learning generalizable visuomotor skills from pre-trained networks." arXiv preprint arXiv:2307.03567 (2023).
>
> [2] Contrastive Learning with Hard Negative Samples : Joshua David Robinson, Ching-Yao Chuang, Suvrit Sra, Stefanie Jegelka
>
> [3] Debiased Contrastive Learning: Ching-Yao Chuang, Joshua Robinson, Lin Yen-Chen, Antonio Torralba, Stefanie Jegelka

---

> > ### Comment · Reviewer_zJpU · 2024-08-08
> >
> > Thank you for running these crucial additional experiments and engaging with my comments in general! I am willing to raise my score.

---

> > > ### Author Response · Authors · 2024-08-08
> > >
> > > Dear Reviewer,
> > >
> > > Thank you so much again for all your feedback that has helped us improve our paper, and for raising your score to "strong accept"!
> > >
> > > Best,
> > > The authors

---

### Official Review · Reviewer_ZVD8 · 2024-07-09

**Soundness:** 2
**Presentation:** 4
**Contribution:** 2
**Rating:** 6
**Confidence:** 4

**Summary:**

This paper studies how diffusion models can be adapted to provide rich representations for training policies. They study how to extract features from a pre-trained Stable Diffusion model in terms of 3 questions: which layers, which diffusion timestep and which text prompts help for downstream task performance. They evaluate these learned representations on a number of robotics tasks like manipulation, navigation and a combination of the two tasks. Their proposed approach performs better than baselines for many tasks.

**Strengths:**

1. Lots of ablation experiments to validate the final design choice.
2. Evaluating a single model for both navigation and manipulation tasks. This is an interesting setup that is becoming more and more relevant with full-body control and robots that need to do both tasks.

**Weaknesses:**

1. It is difficult to ascertain what is the key reason why the performance difference exists between the baselines and proposed method. The baselines have different training data, model sizes, and training losses. While the presented approach has better performance, it is important to identify what is key cause of this: more data, bigger model or the diffusion loss?

Furthermore, there seems to be evidence that for manipulation tasks the baselines (LIV, R3M etc) are good but the performance drops only for navigation tasks which is not the key focus of those baselines.

**Questions:**

1. How do the methods compare when a common model, feature pooling strategy and training dataset are used? It is okay to limit to one kind of task (manipulation/navigation) for this experiment.

2.What if the same feature pooling strategy is applied to baselines like CLIP?. For example if you aggregate spatial features of CLIP from different layers does that help performance? Seems some multi-layer feature aggregation was done for Tables 1(b) and 1(c) but not 1(a) ?

3. Why is CLIP-B used in the ImageNav experiments and not CLIP-L?

**Limitations:**

Yes.

---

> ### Author Rebuttal · Authors · 2024-08-07
>
> We thank you for engaging with our work and commending our presentation, comprehensive ablations and the key result of being able to employ a single representation model for navigation and manipulation tasks. We address your questions and concerns below.
>
> ---
>
> **1) How do the methods compare when a common model, feature pooling strategy and training dataset are used? It is okay to limit to one kind of task (manipulation/navigation) for this experiment.**
>
> Pre-training a foundational representation model is out of the scope of this work and would not be feasible given the limited compute resources available to us. We note that navigation and manipulation tasks are not the primary focus of diffusion model pre-training (similarly to how navigation is not the key focus of baselines that have been designed for manipulation tasks). This is what makes their all-round performance surprising.
>
> **2) What if the same feature pooling strategy is applied to baselines like CLIP?**
>
> **We present multi-layer feature aggregation results for CLIP for all benchmarks in the general response of our rebuttal.** Note that using multiple layers makes the representation size of CLIP twice as large as all the other methods, and is not feasible to do comprehensively for the bigger navigation domains, so we present a smaller ablation there. On a high level, our conclusions from this experiment are that we do see a benefit from including additional outputs from certain layers along with the last layer which is always used (as we anticipated in the discussion in sec 5.1). We find these layers using a grid search and they are located towards the middle (10-14th layers). We note that **while the feature map sizes after including another layer for CLIP are very large and not comparable anymore to the other methods, the performance still doesn’t match that of SCR**.
>
> **3) Seems some multi-layer feature aggregation was done for Tables 1(b) and 1(c) but not 1(a)**
>
> To clarify, we used the same representation extraction procedure for SCR in all of the tables (1a, 1b and 1c), i.e., using the mid+downsampling layers for SCR.
>
> **4) Why is CLIP-B used in the ImageNav experiments and not CLIP-L?**
>
> This is because we compare directly to the results from VC-1 paper [27] where they used the CLIP ViT-base model. We note that ImageNav training runs require 16 GPUs for each run over 6 days of training. We can aim to include a result with CLIP-L in the revised manuscript.
>
> ---
>
> We hope that these clarifications and additional experimental results have addressed any remaining questions and concerns.
>
> If you find that we have addressed your questions and concerns, we kindly ask you to consider raising your score to reflect that. If issues still remain, then we would be more than happy to answer these in the discussion period. Thank you.

---

> > ### Comment · Reviewer_ZVD8 · 2024-08-09
> >
> > Thanks for adding the new experiments with CLIP and answering my questions in detail. Based on the replies, I am updating my score to weak accept.
> >
> > One final comment: given the new CLIP results it might be too harsh to say 'contrastively trained representations such as in CLIP have been shown to fail at enabling embodied agents' in the abstract.

---

> > > ### Author Response · Authors · 2024-08-10
> > >
> > > Dear Reviewer,
> > >
> > > Thank you so much for your feedback which has helped us improve the paper and for recommending acceptance!
> > >
> > > We will change the phrasing as you suggested and include a subsection to discuss these findings in the revised manuscript!
> > >
> > > Best,
> > > The authors

---

### Official Review · Reviewer_sdTb · 2024-07-10

**Soundness:** 3
**Presentation:** 3
**Contribution:** 2
**Rating:** 6
**Confidence:** 3

**Summary:**

This paper introduces SCR, a method that extracts representations from a pre-trained text-to-image diffusion model for learning downstream robotics control policies.

Given a noised input image and prompt text, the visual representation is extracted from selected layers of the denoising U-Net. The extraction method is meticulously designed with extensive experiments and analysis to support the design choices.

Experiments on multiple benchmarks across three tasks—few-shot imitation learning, image-goal navigation, and open-vocabulary mobile manipulation—demonstrate competitive performance, verifying the method’s ability to generalize to open-ended environments.

**Strengths:**

1. Although the idea of replacing CLIP with diffusion models for extracting representations for robotic control is straightforward, the technical details, supported by comprehensive ablation and analysis of the design choices, are insightful and convincing.

2. The paper has potential for future impact. The method's ability to handle complex and open-ended environments suggests it could be generalized and applied to many other tasks.

3. The paper is well-written and easy to read.

4. The comparison experiments are thorough across multiple tasks and benchmarks, demonstrating the proposed method's strong performance.

**Weaknesses:**

1. As noted in the related work, [50] found that the optimal noising timestep selection can vary depending on the task due to the required granularity of predictions. However, the ablation study for fixed timestep selection is performed only on the Franka-Kitchen benchmark. Would the results differ if tested on different tasks or benchmarks?

2. Additionally, all ablation studies in Section 5 are conducted solely on the Franka-Kitchen benchmark. SCR-FT shows a performance drop compared to the non-finetuned version on the ImageNav benchmark, attributed to the finetuning being performed only with table-top manipulation datasets. This raises concerns that different tasks have notable domain gaps and may require different features from the inputs. More experiments, or at least reasonable explanations, on whether the optimal framework design varies across tasks would be beneficial.

**Questions:**

Please kindly refer to the weaknesses.

**Limitations:**

Yes, the authors discussed the limitations that downstream performance depends on design choices for representation extraction and the higher run-time cost of using a pre-trained model with more parameters.

---

> ### Author Rebuttal · Authors · 2024-08-07
>
> Thank you for taking the time to review our manuscript and for providing helpful and constructive feedback. We were happy to see your review note our study’s potential for future impact, the thoroughness of our experiments and ablations which demonstrate the strong performance of the approach we study, and the clarity of presentation. We address your concerns below.
>
> ---
>
> **1) [50] found that the optimal noising timestep selection can vary depending on the task .. the ablation study for timestep is performed only on the Franka-Kitchen benchmark. Would results differ if tested on different tasks?**
>
> Yes, our observation is consistent with prior work. **The optimal noising timestep depends on the task** and we present the same ablation for timestep on the Meta-World environment below. SCR performs similarly across 0, 100 and 200 timestep on Meta-World because the images on this benchmark contain coarser details. We have provided further intuition and discussion around how to choose the best noising timestep in Figure 8 in the appendix, where we show reconstructions obtained from the diffusion model for different benchmark images starting at different denoising timesteps.
>
> | Timestep | Success       |
> | -------- | ------------- |
> | 0        | 94.1 &pm; 1.9 |
> | 0        | 94.4 &pm; 1.9 |
> | 0        | 94.4 &pm; 2.0 |
>
> **2) All ablation studies in Section 5 are conducted solely on the Franka-Kitchen benchmark.**
>
> Thank you for suggesting this, **we conducted additional ablations on the Meta-World benchmark and present these in the general response of the rebuttal**. Note that ablating the layer + noise selection on Meta-World shows that we can beat the best method there (R3M[30]) if we use a smaller subset of the layers we used (thereby achieving a 97% success rate), however we recommend retaining the standardized settings which we used across benchmarks to retain simplicity while still getting overall good performance across tasks. Since the navigation benchmarks require a lot of compute resources, we can provide a full set of ablations on these for the camera ready version, if the reviewer suggests it.
>
> **3) SCR-FT shows a performance drop compared to the non-finetuned version on the ImageNav benchmark.**
>
> We agree that a visual domain gap exists between the manipulation and navigation benchmarks, which **could be mitigated by co-finetuning with manipulation and navigation datasets (like Real Estate 10k[1])** as done in VC-1[27].
> With SCR-FT, we aimed to present a proof-of-concept result for the simple non-task-specific fine-tuning that is possible when using an image generation representation model, using some of the commonly-used datasets in the representation learning literature - most of which focus on few-shot manipulation tasks. We also presented a negative result of trying to do this for the CLIP model in Appendix D.1.
> Finally, **we have aimed to keep the representation selection procedure standardized for all tasks**- by deciding a common subset of layer outputs which we use as the representation, as well as the denoising timestep value of 0 since that always does well (while higher denoising timesteps might also do well on some benchmarks). Beyond this there are no additional hyperparameters introduced for SCR.
>
> ---
>
> We hope that these clarifications and additional experimental results have addressed any remaining questions and concerns.
>
> If you find that we have addressed your questions and concerns, we kindly ask you to consider raising your score to reflect that. If issues still remain, then we would be more than happy to answer these in the discussion period. Thank you.
>
> ---
>
> **References**
>
> [1] Zhou, Tinghui, et al. "Stereo magnification: Learning view synthesis using multiplane images." arXiv preprint arXiv:1805.09817 (2018).

---

> > ### Comment · Reviewer_sdTb · 2024-08-11
> >
> > Thanks for the clarifications and the additional experiments. They have answered my questions. I believe the original score accurately reflects my positive assessment of the paper.
> >
> > I noticed a small typo in the table for question 1—the first column says "0, 0, 0", which I assume should be "0, 100, 200" given the texts. This doesn't impact my overall assessment.

---

### Official Review · Reviewer_QYDZ · 2024-07-12

**Soundness:** 2
**Presentation:** 4
**Contribution:** 3
**Rating:** 6
**Confidence:** 4

**Summary:**

This paper introduces Stable Control Representations (SCR), a novel approach that aggregates intermediate outputs from diffusion models for robotic control tasks. The authors validate its effectiveness on various benchmarks, including manipulation, navigation, and grasp point prediction. The key design space has been thoroughly ablated, showing SCR's superiority over other visual representation methods.

**Strengths:**

1. Innovative Exploration: Visual representation is a fundamental challenge in robotics. Exploring the effectiveness of diffusion models in this context is intriguing and novel.
2. Comprehensive Experiments: The authors conducted extensive experiments covering various robotic tasks, diffusion steps, intermediate layer selection, and aggregation methods. The design space has been thoroughly explored.
3. Clear Writing and Organization: The paper is well-written and organized, making the narrative easy to follow.

**Weaknesses:**

1. Lack of Real-World Validation: While SCR performs well on simulation baselines, real-world validation is missing. Demonstrating its effectiveness on real robots would significantly enhance the credibility and practical applicability of SCR.
2. Insufficient Evidence: Although the authors provide extensive empirical evidence of SCR's effectiveness, the paper lacks theoretical and quantitative analysis explaining why diffusion-based visual representations outperform other baselines. This theoretical backing would strengthen the paper's overall argument.

**Questions:**

None

**Limitations:**

Refer to the weaknesses section.

---

> ### Author Rebuttal · Authors · 2024-08-07
>
> Thank you for taking the time to review our manuscript and for providing helpful and constructive feedback. We were happy to see your review note our study’s innovative exploration, thoroughness of our experiments which demonstrate the strong performance of the approach we study, and the clarity of presentation. We address your concerns below.
>
> ---
>
> **1) Real-world validation**: We employed a broad range of tasks in our work to add evaluation signals from different domains (manipulation and highly photorealistic indoor navigation) and learning setups (few-shot vs RL). Real robot experiments require an expensive setup and largely feature in works that look at few-shot table-top manipulation tasks which can provide noisy performance signals due to low sample sizes. We believe this would not have added much additional value to our exploration of simply validating the usability of diffusion model representations as a first step, and is better suited for future work. Many influential works in this space also rely completely on evaluation in simulation environments [1, 2, 3, 4].
>
> **2) Theoretical evidence**: Consistent with the embodied AI literature (e.g., VC-1[27], R3M[30], Voltron [20] etc.), our work focuses on displaying strong empirical evidence for the usefulness of pretrained representations for a wide range of control tasks. We agree that studying the properties of the representation manifold of denoising diffusion models is an interesting and impactful direction for future work.
>
> ---
>
> We hope that these clarifications have addressed any remaining questions and concerns.
>
> If you find that we have addressed your questions and concerns, we kindly ask you to consider raising your score to reflect that. If issues still remain, then we would be more than happy to answer these in the discussion period. Thank you.
>
> ---
>
> **References**
>
> [1] Khandelwal, Apoorv, et al. "Simple but effective: Clip embeddings for embodied ai." Proceedings of the IEEE/CVF Conference on Computer Vision and Pattern Recognition. 2022.
>
> [2] Xiao, Tete, et al. "Masked visual pre-training for motor control." arXiv preprint arXiv:2203.06173 (2022).
>
> [3] Yadav, Karmesh, et al. "Offline visual representation learning for embodied navigation." Workshop on Reincarnating Reinforcement Learning at ICLR 2023. 2023.
>
> [4] Eftekhar, Ainaz, et al. "Selective visual representations improve convergence and generalization for embodied ai." arXiv preprint arXiv:2311.04193 (2023).

---

> > ### Comment · Reviewer_QYDZ · 2024-08-10
> >
> > Thanks for your response. I agree that real-world experiments are unnecessary at this stage and I'm willing to raise the score.

---

> > > ### Author Response · Authors · 2024-08-10
> > >
> > > Dear Reviewer,
> > >
> > > Thank you again for your valuable feedback that has helped us improve our paper, and for raising your score!
> > >
> > > Best,
> > > The authors

---

### Official Review · Reviewer_9EUL · 2024-07-15

**Soundness:** 3
**Presentation:** 3
**Contribution:** 3
**Rating:** 7
**Confidence:** 3

**Summary:**

The paper investigates whether latent representations from a pre-trained tex-to-image generation model are useful for object manipulation or navigation tasks in embodied agents. The paper considers a few design choices in how to extract latent features from Stable Diffusion, that could lead to more versatile and useful features for downstream control tasks. The method is referred to as Stable Control Representation (SCR) with variants like additionally using cross-attention maps (SCR-Attn) or a finetuned version (SCR-FT). Experimental results show that SCR series can achieve comparable or better performance than general VL encoders like CLIP or method designed for control tasks.

**Strengths:**

- The paper studies an interesting problem of using text-to-image diffusion models for embodied agent control. The paper is well-written and easy to understand.
- The method considers many design choices of SCR in a very comprehensive way. The choices of different factors, from denoising steps to layer selection, have been clearly presented and justified later in the experiment.
- The experiment seems well-designed and comprehensive. Most of the results support the benefit of using SCR for control tasks.

**Weaknesses:**

- The method is loosely connected with diffusion mechanism and simply adopt Stable Diffusion as a feature extractor with one pass if I understand correctly. This loose connection is particularly amplified in Table 2a where timestep=0 ends up with the best performance. This basically indicates that the "generative" power of text-to-image models are not necessary for the task. In that sense, is it still necessary to use T2I diffusion models if one could find a visual encoder (e.g. ViT) trained with similar amount of images as Stable Diffusion?
- The performance gap between SCR-FT and R3M in Meta-World and Franka-Kitchen indicates that the method may not generalize well to some scenarios. Could you give any insights about the performance gap? Is it related to domain shift or image style differences or any other reaons?

- The order of Table 2 is different from descriptions in Sec. 5.1 and 5.2

**Questions:**

See above

---

> ### Author Rebuttal · Authors · 2024-08-07
>
> Thank you for taking the time to review our manuscript and for providing helpful and constructive feedback. We were happy to see your review mention the clarity of our presentation, the relevance of the problem we are studying, and the comprehensiveness of our evaluation. We address your concerns below.
>
> ---
>
> **1) Can a visual encoder (like ViT) trained on a similar dataset as Stable Diffusion suffice, as the generative aspect of T2I diffusion models does not seem to be essential for the task with t=0.**
>
> Tackling complex open-ended embodied tasks requires foundation vision-language model representations. We study the Stable Diffusion model since other existing foundation models like CLIP have their own shortcomings, we expand on this below.
> - The foundation model representations which have been considered for control so far are trained with auto-encoding (MAE) or contrastive learning objectives (CLIP[33], LIV[25], R3M[30]) - and prior work [20] has noted that these seem to do well on disparate sets of tasks depending on the level of visual and semantic details they encode (with image-text contrastive approaches encoding semantics better and auto-encoding approaches encoding more finer details). **We hypothesized that the text-conditioning of a T2I generative model might enforce encoding semantics and the generation objective might incentivise retention of more fine grained visual detail in the representations**. The question was then whether it would be possible to localize a subset of representations that would sufficiently encode these details and do well across tasks. This is the main result we demonstrate.
> - Moving to language-conditioned control tasks necessitates investigating vision-language model representations. Given the limited effectiveness of CLIP (which is the primary candidate for a vision language representation) as a backbone on control benchmarks, we intended our work to be an explorative study of a different kind of foundation model that could provide vision-language aligned representations for control. Since there is a **lot of research momentum in the direction of image and video generation diffusion models**, successfully leveraging them for control would be beneficial in the long run.
>
> **2) This loose connection is particularly amplified in Table 2a where timestep=0 ends up with the best performance. This basically indicates that the "generative" power of text-to-image models is not necessary for the task.**
>
> - We note that when using a noise value of 0, the U-net still forms a meaningful representation of the input. [2] also uses zero noising for input images, to do zero-shot open-set segmentation using intermediate outputs from the U-Net. [3] also follow the score estimates from the U-net at t=0 to continue to keep refining the data samples and to bootstrap a Langevin sampler from that. The fact that the score function gradient estimates are informative at t=0 implies that the intermediate representations in the model are informative too.
> - We also note that the MAE (Masked AutoEncoders) [1] representation model (which is the backbone for the VC-1 model) follows a similar strategy. It is trained to reconstruct missing patches from the input, but when it is used as a representation backbone none of the patches are masked out. The uncorrupted input is passed through the model to derive the representation, thereby not exactly using the denoising power of the MAE in the forward pass.
>
> **3) Explain The performance gap between SCR-FT and R3M in Meta-World and Franka-Kitchen**
> We are happy to provide more context about performance differences on the benchmarks in the revised manuscript. R3M representations are highly sparse and tailored specifically to reduce overfitting in few-shot IL settings (see Table 1 in the R3M paper where performance drops 4-6% when the L1 sparsity penalty is removed), which does not end up being an advantage on other kinds of tasks with large scale RL training. We also note that we standardized representation extraction settings across all benchmarks for SCR for simplicity and reproducibility, but if we had tuned the layer selection choice for each of the domains specifically, we would be able to beat R3M on Meta-World with the setting where we use the mid+down_3 layers with t=200 timestep, and get 97% on Meta-World (this result is presented in the general response of our rebuttal, where we replicate our Franka ablations for the layer and noise parameters on Meta-World).
>
> **4) Typos**
>
> Thank you for pointing out the swapped order of table column references in sections 5.1 and 5.2, we have fixed this in the manuscript.
>
> ---
>
> We hope that these clarifications have addressed any remaining questions and concerns.
>
> If you find that we have addressed your questions and concerns, we kindly ask you to consider raising your score to reflect that. If issues still remain, then we would be more than happy to answer these in the discussion period. Thank you.
>
> ---
>
> **References**
>
> [1] He, Kaiming, et al. "Masked autoencoders are scalable vision learners." Proceedings of the IEEE/CVF conference on computer vision and pattern recognition. 2022.
>
> [2] Open-Vocabulary Panoptic Segmentation with Text-to-Image Diffusion Models. Xu et al., 2023.
>
> [3] Iterated Denoising Energy Matching for Sampling from Boltzmann Densities. Akhound-Sadegh et al., 2024.

---

> > ### Author Response · Authors · 2024-08-12
> >
> > Dear Reviewer 9EUL,
> >
> >
> > The discussion periods ends tomorrow (August 13).
> >
> > We have made considerable effort to address the concerns and queries you have raised, and would be grateful for the opportunity to clear up any remaining queries you may have while we are still in the discussion period.
> >
> >
> > Thank you.

---

> > ### Comment · Reviewer_9EUL · 2024-08-12
> > **Thank you for the response**
> >
> > Most of the responses are convincing and have addressed my concerns. But I am not sure if I understand the sentence, *We hypothesized that the text-conditioning of a T2I generative model might enforce encoding semantics and the generation objective might incentivise retention of more fine grained visual detail in the representations.*, correctly. Is it because that SD adopts a freezed CLIP text encoder, so that you hypothesize that the text encoder enforces encoding semantics? (Btw, what is enforced to encode semantics by the text-conditioning stage?) I guess I just don't see how the concept of "encoding" or "retention of visual details" could fit into the diffusion process or is it just a very high-level interpretation as the target is to extract representations from a generative model?

---

> > > ### Author Response · Authors · 2024-08-13
> > >
> > > Thank you for your response! We are happy to clarify our statement with respect to your question.
> > >
> > > > “I am not sure if I understand the sentence, “We hypothesized that the text-conditioning of a T2I generative model might enforce encoding semantics and the generation objective might incentivise retention of more fine grained visual detail in the representations., correctly. "
> > >
> > > > "Is it because that SD adopts a freezed CLIP text encoder, so that you hypothesize that the text encoder enforces encoding semantics? (Btw, what is enforced to encode semantics by the text-conditioning stage?)”
> > >
> > > Yes, more specifically, the signal from the text prompts is incorporated into the U-Net through the use of a (dot-product-based) cross-attention layer within each block. This incentivises the visual feature maps in each block to align with the text encodings selectively, thereby also aligning with language concepts. This is also evidenced by the fact that we can isolate word-aligned attention maps for each U-Net layer, which roughly correspond to the entity in the image referenced by the word (when projected onto the image). This is shown in Figures 9 and 10 of the appendix in our submission.
> > >
> > > > “I guess I just don't see how the concept of "encoding" or "retention of visual details" could fit into the diffusion process or is it just a very high-level interpretation as the target is to extract representations from a generative model?”
> > >
> > > Diffusion-based image generation models are trained to generate a final image by producing progressively more denoised versions of an existing noised image. To be able to refine and produce the image corresponding to the last few steps of denoising (likely only needing a few pixels to be edited or denoised), the model needs to encode the same level of visual details that a reconstruction/auto-encoding model might also encode. The works [55, 57] which we cite within our related work section also provide similar evidence for representation learning within diffusion models (for computer vision tasks). We also refer the reviewer to a recent survey paper that explores the interplay between diffusion models and representation learning in more detail.
> > >
> > > Thank you for giving us the opportunity to clarify the points above. Please let us know if you have any further questions!
> > >
> > > [1] Diffusion Models and Representation Learning: A Survey, https://arxiv.org/abs/2407.00783

---

> > > > ### Comment · Reviewer_9EUL · 2024-08-13
> > > >
> > > > Thanks for the detailed response with references. It's interesting to learn about the connections and see how SD is also useful in embodied controls. I have raised my score.

---

> > > > > ### Author Response · Authors · 2024-08-13
> > > > >
> > > > > Dear Reviewer,
> > > > >
> > > > > Thank you again for your engagement and valuable feedback that has helped us improve our paper, and for raising your score!
> > > > >
> > > > > Best,
> > > > >
> > > > > The authors

---

### Author Rebuttal · Authors · 2024-08-07

We thank the reviewers for taking the time to review our manuscript and for providing thoughtful and constructive feedback.

We were delighted that reviewers recognized that our work studies an “important” and “interesting” problem (**9EUL, QYDZ**) while being presented in a manner which is “easy to understand” (**9EUL**) and which “contextualizes the background and related work well” (**zJpU**). We are pleased to see the reviewers note the “strong performance” (**sdTb**) of the proposed approach and appreciate the “thorough ablations and analysis” (**QYDZ, sdTb**), along with a “fair treatment of the baselines” (**zJpU**) which we strived to ensure in our work.

Reviewers also commended the “comprehensive evaluation” (**all**), "which becomes more important with paradigm shift towards multi task models" (**ZVD8**). We are grateful that they point out the work has “potential for future impact through the ability to handle complex open-ended environments” (**sdTb**). Finally, we thank the reviewers for acknowledging our open source code repository which will help reproducibility and accelerate future efforts in this direction (**zJpU**).

---

**Since reviewers ZVD8 and zJpU asked for results on multi-layer aggregation of CLIP representations, we present these results in the general response:**

As we note in Section 5.1 of the manuscript, we can see clear benefits from aggregating multi-layer features for SCR. The reason why we used these features is because of the smaller feature maps in the U-Net compared to ViTs (thus using multiple layer outputs for a fair comparison). This is an architecture-specific property, and not a property of diffusion models specifically, and we expect all models to benefit from this. However, as we mention in L262, this can lead to very large representation sizes for ViTs (each layer output is 16x16x1024).

Following the reviewer’s suggestion, we ran the same experiment for CLIP by concatenating outputs from the last layer + another layer in the range of layers 10-2.1 (We use two layers to keep GPU memory manageable, and note that this makes the CLIP representations twice as large compared to other methods.) We present the results below:

| Model     | Layers  		   | Success (&pm; std err)  |
| --------- | ---------------- | ----------------------- |
| CLIP-L    | 23 (last layer)  | 36.3 &pm; 1.7           |
| CLIP-L    | 21+23 		   | 35.4 &pm; 2.9           |
| CLIP-L    | 19+23 		   | 38.5 &pm; 3.2           |
| CLIP-L    | 17+23 		   | 39.0 &pm; 3.0           |
| CLIP-L    | 12+23 		   | 40.8 &pm; 2.8           |
| CLIP-L    | 10+23 		   | 40.2 &pm; 3.2           |
| SCR (ours)| Down[1-3] + Mid  | **49.9 &pm; 3.4**       |

We see an interesting trend, namely that moving towards middle layers leads to higher performance indicating that CLIP layers around layer 10-14 encode some details useful to the Franka benchmark. **While we do see a benefit from including the output of certain additional layers, performance CLIP performance still does not match SCR.**
We also wanted to see if the best pair of layers that we found here might be better on other tasks as well. To test this, we ran the same combination for Meta-World, OVMM, and Imagenav (adjusted for ViT-B). We present the results below.

| Model     | Layers  		  | Meta-World        | OVMM          |
| --------- |---------------- | -------------     | ------------- |
| CLIP-L    | 23 (Last Layer) | 90.1 &pm; 3.6     | 38.7 &pm; 1.7 |
| CLIP-L    | 12+23           | 91.7 &pm; 2.6     | 38.6          |
| CLIP-L    | 21+23           | 91.2 &pm; 2.3     | -             |
| SCR (ours)| Down[1-3] + Mid | **94.9 &pm; 2.0** | **43.6 &pm; 2.1** |

| Model     | Layers  		  | ImageNav |
| --------- |---------------- | -------- |
| CLIP-B    | 11 (Last Layer) | 52.2     |
| CLIP-B    | 6+11            | 66.6     |
| SCR (ours)| Down[1-3] + Mid | **73.9** |


We see a significant improvement on ImageNav and a slight improvement on Meta-World. (We did not see an improvement on OVMM but note that the results above are only for a single seed.)

---

**Following Reviewer sdTb's suggestion, we also replicated our ablations---which we had previously performed on the Franka kitchen benchmark---on Meta world. We present the results here**:

**Meta-World Ablations**
| Layers          | Noise | Success (&pm; std err) |
| --------------- | ----- | ---------------------- |
| Mid             | 200   | 94.7 ± 2.8             |
| Down[3] + Mid   | 200   | **97.3 ± 1.4**         |
| Down[1-3]       | 200   | 94.1 ± 1.9             |
| Down[1-3] + Mid | 200   | 94.4 ± 1.9             |
| Down[1-3] + Mid | 100   | 94.4 ± 1.9             |
| Down[1-3] + Mid | 0     | 94.1 ± 1.9             |

The top three rows ablate the layer used, and the bottom three rows ablate the noise values. We note that this ablation on Meta-World shows that it is possible to outperform the best method here (R3M[30]) if we use a smaller subset of the layers we used as the default (thereby achieving a 97% success rate). However, standardizing the representation extraction settings which we used across benchmarks helps us to retain simplicity while still achieving an overall good performance across tasks.

---

### Decision · Program_Chairs · 2024-09-25

**Decision:**

Accept (spotlight)

**Comment:**

The work introduces Stable Control Representations, a technique that utilizes representations from general-purpose, pre-trained diffusion models for control tasks and are capable of capturing both the abstract high-level and fundamental low-level details of a scene. Experiments demonstrate improvements in generalization across various tasks.

Summary Of Reasons To Publish:

1) Novel approach proposing diffusion Model representations work for robotic control tasks.
2) Well-motivated and well-structured paper
3) Solid experiment setup and results

Summary Of Suggested Revisions:

No major revisions are suggested by the reviewers. I suggest authors to fix the presentation issues raised by reviewers in the final version.